# PB²: Preference Space Exploration via Population-Based Methods in Preference-Based Reinforcement Learning

## Abstract

Preference-based reinforcement learning (PbRL) has emerged as a promising approach for learning behaviors from human feedback without predefined reward functions. However, current PbRL methods face a critical challenge in effectively exploring the preference space, often converging prematurely to suboptimal policies that satisfy only a narrow subset of human preferences. In this work, we identify and address this preference exploration problem through population-based methods. We demonstrate that maintaining a diverse population of agents enables a more comprehensive exploration of the preference landscape compared to existing single-agent approaches. Crucially, this diversity improves reward model learning and leads to preference queries with clearly distinguishable behaviors that are less ambiguous to the user. Our experiments reveal that current methods may get stuck in local optima and require an excessive amount of feedback, or may degrade significantly in performance when human evaluators make errors on similar trajectories—a realistic scenario often overlooked by methods relying on perfect oracle teachers. In contrast, our population-based approach has enhanced preference exploration capabilities in environments with complex reward landscapes, and is significantly more robust to noisy teachers.

## 1 Introduction

Reinforcement learning (RL) has demonstrated remarkable success across a wide range of applications, from game playing to robotic control. However, the effectiveness of traditional RL methods remains heavily dependent on carefully designed reward functions, which are often challenging to specify for complex tasks involving subjective outcomes or intricate human preferences (Hadfield-Menell et al., 2017). Preference-based reinforcement learning (PbRL) offers a promising alternative by enabling agents to learn directly from human feedback through preferences between pairs of behavior trajectories (Christiano et al., 2017; Lee et al., 2021a). This approach eliminates the need for hand-crafted reward functions and provides a more intuitive interface for humans to express their intentions. Despite these advantages, PbRL faces a fundamental challenge: policies optimized for the current reward model often generate similar queries for preference elicitation, limiting the informativeness of human feedback needed to improve the model.

Existing approaches attempt to address preference elicitation challenges through various query selection strategies: uncertainty-based methods targeting uncertain preference regions (Marta et al., 2023), policy-aligned techniques focusing on current behavior (Hu et al., 2024), ensemble-based strategies leveraging model disagreement (Liang et al., 2022), and information-theoretic approaches maximizing information gain (Biyik et al., 2020; Biyik and Sadigh, 2018). While improving sample efficiency, these methods operate within a single-agent framework, limiting behavioral diversity during evaluation. This becomes particularly problematic when humans must compare similar trajectories, as evaluators often provide inconsistent feedback in such cases (Huang et al., 2025), adding noise that significantly degrades learning performance. Moreover, one of the main claims of this paper is that existing reward modeling approaches based on neural ensembles (Christiano et al., 2017; Lee et al., 2021b; Liang et al., 2022) induce in reality little behavioral diversity and might be a poor representation of the rewards' posterior distribution. As such, without an explicit mechanism for maintaining diversity, existing approaches remain vulnerable to local optima—requiring excessive

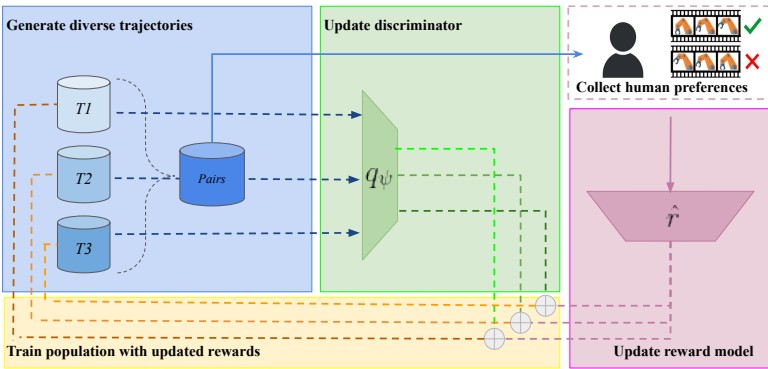

Figure 1: **Overview of PB²**. Agents in the population optimize a combination of a reward learned from user queries and a bonus term from a discriminator trained to maintain agent distinguishability. This combined reward encourages the discovery of diverse behaviors, while maintaining alignment with human preferences, thereby helping to propose informative and less ambiguous user queries.

feedback to recover—and have severely degrading performance under realistic conditions of human evaluation inconsistency.

To improve on this lack of behavioral diversity during the user elicitation step of PbRL, we present in this paper PB²: a novel population-based approach to preference space exploration. Our key insight is that simultaneously training multiple distinct policies with an explicit diversity bonus facilitates more thorough preference landscape exploration than conventional single-policy methods. By collecting experiences across different policies to construct comparison pairs, we substantially enhance the variety of behaviors evaluated while preserving alignment with expressed human preferences. As illustrated in Fig.1, our approach implements this insight through a feedback loop where diverse policies generate distinct trajectories, human preferences on these trajectories train a reward model, and a discriminator maintains population diversity while encouraging behaviors that align with current preferences. Our contributions are as follows:

1. We identify the preference exploration problem in PbRL, demonstrating how single-agent methods frequently fail by converging to suboptimal local minima in the preference space.

2. We propose a population-based framework for PbRL that maintains policy diversity while optimizing for human preferences, achieving a balance between exploration and exploitation while significantly enhancing exploration of the preference landscape.

3. We demonstrate through experiments that PB² produces more distinguishable queries that improve reward learning efficiency, achieving greater robustness when human feedback is inconsistent and improving performance with limited feedback.

4. We demonstrate the limitations of neural ensemble modeling for capturing the uncertainty over user preferences, showing experimentally that it behaves almost the same as not modeling reward uncertainty at all.

We validate these claims through three complementary experimental evaluations: (1) a systematic evaluation across DMControl locomotion tasks with varying similarity thresholds $\epsilon$ to simulate human judgment inconsistency, and (2) a qualitative demonstration of how PB² escapes local optima in complex preference landscapes where single-agent methods remain trapped and (3) a comparative analysis in navigation tasks with extremely limited feedback, demonstrating PB²'s feedback efficiency.

## 2 RELATED WORK

**Preference-based Reinforcement Learning.** Preference-based Reinforcement Learning (PbRL) enables agents to learn from human feedback through trajectory comparisons (Christiano et al., 2017; Lee et al., 2021a; Wirth et al., 2017). Despite its intuitive interface, PbRL faces a fundamental exploration challenge, as policies optimized for the current reward model often generate similar queries for preference elicitation, limiting the informativeness of human feedback needed to improve the

model. Recent methods address this through reward uncertainty (Liang et al., 2022), semi-supervised learning (Park et al., 2022), model-based approaches (Liu et al., 2023), bi-level optimization (Liu et al., 2022), dynamics encoding (Metcalf et al., 2022), and query-policy alignment (QPA, (Hu et al., 2024)), though they remain primarily exploitative. RIME (Cheng et al., 2024) addresses a complementary challenge of noisy feedback through learning to detect and handle mislabeling errors.

Recent work has also explored different strategies for improving query selection and exploration. PPE (Zhu et al., 2024) improves preference buffer coverage through proximal policy exploration guided by out-of-distribution detection and a mixture distribution query method that balances exploration of new trajectories with maintaining reward model accuracy on in-distribution data. SENIOR (Ni et al., 2025) combines motion-distinction-based query selection with preference-guided exploration using intrinsic rewards, though operating in offline settings. While these methods improve various aspects of PbRL, they operate within single-agent frameworks that limit behavioral diversity during query generation.

Bayesian approaches like Dueling Posterior Sampling (DPS, Novoseller et al. (2020)) generate queries from policies maximizing posterior samples of the reward function, providing strong theoretical guarantees for tackling the exploration/exploitation of user preferences. Our approach can be thought of as an instantiation of DPS, but we show in this paper that a naive implementation thereof without an explicit diversity bonus is in fact not much different than the purely exploitative QPA algorithm that only optimizes the maximum likelihood reward with no attempt at reward uncertainty modeling.

**Population-based Reinforcement Learning and Quality-Diversity.** Population-based methods maintain multiple agents to enhance exploration, from Population-Based Training (Jaderberg et al., 2017) to extensions incorporating Bayesian optimization (Wan et al., 2022), long-term performance (Dalibard and Jaderberg, 2021), and evolutionary selection (Salimans et al., 2017; Alam et al., 2020). Quality Diversity algorithms discover diverse high-performing solutions, evolving from Novelty Search (Lehman, 2012) through approaches like NSLC (Lehman and Stanley, 2011) and MAP-Elites (Mouret and Clune, 2015), with recent integration into RL (Nilsson and Cully, 2021; Cully, 2019). Our approach shares core principles with these methods but targets preference landscape exploration rather than hyperparameter optimization, building on diversity-promoting concepts like DvD (Parker-Holder et al., 2020) while uniquely aligning diversity with human preferences. Unlike QD methods requiring manually designed descriptors, our method automatically identifies meaningful behavioral patterns while adapting to evolving human preferences rather than pursuing general diversity.

**Unsupervised Skill Discovery.** Skill discovery leverages intrinsic motivation through mutual information maximization (Eysenbach et al., 2019), dynamics-awareness (Sharma et al., 2020), and task-aligned methods (Kumar et al., 2020; Osa et al., 2022). Our approach is inspired by SMERL (Kumar et al., 2020) but operates without expert access, encouraging distinctiveness between behaviors from different population members rather than between latent-conditioned policies of a single agent. However, despite these advances in diversity promotion across different domains, these methods have yet to make their way into PbRL literature to maintain behavioral diversity during query generation, which we show to be a challenge for current methods in Sec. 4.

## 3 PRELIMINARIES

This section establishes the theoretical foundations for our approach, beginning with standard RL concepts before introducing preference-based learning and the challenges it faces.

**Reinforcement Learning.** We consider the standard reinforcement learning framework with an environment modeled as a Markov Decision Process (MDP) $M := (S, A, T, r, \gamma)$ (Sutton et al., 1998), where $S$ is the state space, $A$ is the action space, $T$ is the transition function, $r$ is the reward function, and $\gamma$ is the discount factor. A trajectory $\tau = (s_0, a_0, s_1, a_1, \ldots)$ represents a sequence of state-action pairs generated by following policy $\pi$, and the return $R(\tau) = \sum_{t=0}^{\infty} \gamma^t r(s_t, a_t)$ quantifies the total discounted reward obtained along this trajectory. The agent's goal is to find a policy $\pi$ that maximizes the expected discounted cumulative reward, $\mathbb{E}_\pi[\sum_{t=0}^{\infty} \gamma^t r(s_t, a_t)]$.

**Preference-Based Reinforcement Learning.** In preference-based reinforcement learning (PbRL) (Christiano et al., 2017; Wirth et al., 2017), the agent lacks access to an explicit reward function. Instead, the learning process relies on human feedback provided through pairwise comparisons. During training, the algorithm generates queries by selecting pairs of trajectory segments $(\sigma^0, \sigma^1)$

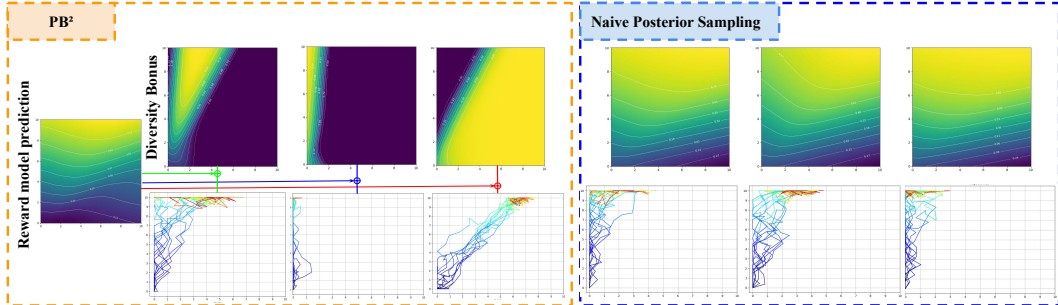

Figure 2: **Diverse population-based query strategy improves preference space coverage.** 2D navigation task with a goal on the top right corner. ***Orange box*** shows the current reward model that prefers states in the upper region, and diversity bonuses (Sec. 5.1) for three agents, with distinct spatial concentrations. Bottom row shows agent trajectories maximizing the combination of the learned reward model and the diversity bonus, except for the first agent that always only maximizes the current reward model in PB². This agent corresponds to what QPA would learn. On itself, this agent only generates a moderate amount of behavioral diversity stemming from its policy entropy, while the population of PB² collectively covers a wider region of the state space while staying aligned with human preferences. This desirable property is not as clear without an explicit diversity bonus as shown in the ***Blue box***. Here agents maximize their individual reward models (top row) of a learned neural ensemble following standard PbRL procedures. While these reward functions would ideally represent samples of the posterior reward distribution, they end up very similar looking and induce trajectories with a lot more overlap than PB², suggesting that existing reward modeling techniques in PbRL fail to properly represent the uncertainty in user preferences.

from the agent's experience, where a segment $\sigma$ is a sequence of state-action pairs extracted from collected trajectories. These segment pairs are presented to a human teacher, who provides preference feedback by indicating which segment better aligns with the desired behavior. The preference label $y$ is a binary indicator: $y^{(1)} = 1$ if the human prefers $\sigma^1$ over $\sigma^0$, and $y^{(0)} = 1$ otherwise.

Following the Bradley-Terry model (Bradley and Terry, 1952; Fürnkranz et al., 2012), we learn a reward function $r_\phi$ that induces a preference predictor:

$$P_\phi[\sigma^1 \succ \sigma^0] = \frac{\exp\left(\sum_{t=1}^{T} \gamma^{t-1} r_\phi(s_t^1, a_t^1)\right)}{\sum_{i \in \{0,1\}} \exp\left(\sum_{t=1}^{T} \gamma^{t-1} r_\phi(s_t^i, a_t^i)\right)} \tag{1}$$

The reward function parameters $\phi$ are optimized by minimizing the cross-entropy loss over collected preference data $D$:

$$L_{\text{reward}} = -\mathbb{E}_{(\sigma^0, \sigma^1, y) \sim D} \left[ y^{(0)} \log P_\phi[\sigma^0 \succ \sigma^1] + y^{(1)} \log P_\phi[\sigma^1 \succ \sigma^0] \right] \tag{2}$$

This learned reward function then guides policy training through standard RL algorithms. The reward function parameters $\phi$ are shared across all agents in the population, providing a common objective while agents maintain distinct behaviors through the diversity mechanism described in Section 5.1. However, the effectiveness of this approach critically depends on how queries are generated and selected.

## 4 BALANCING PREFERENCE SPACE EXPLORATION AND EXPLOITATION

In PbRL, human feedback serves as a crucial but costly resource. The strategic selection of queries presented to human evaluators significantly impacts the learning process. PbRL systems face a fundamental challenge: effectively exploring the preference space to discover human preferences while exploiting current knowledge to improve policies, requiring two competing objectives: **exploitation** which consists in refining the reward model in areas most relevant to current policy improvement and **exploration** that aims to discover previously unknown aspects of human preferences.

Current approaches struggle with this balance. PEBBLE (Lee et al., 2021b) samples queries from outdated trajectories, creating temporal misalignment between feedback and current policies. Uncertainty-driven approaches like RUNE (Liang et al., 2022) often prioritize informationally rich regions that may be irrelevant to current capabilities. Even QPA (Hu et al., 2024), which aligns queries with current policies, becomes overly exploitative, creating blind spots in the preference model and potentially trapping agents in local optima.

Counterintuitively, despite extensive research into active query selection, the current PbRL state-of-the-art (QPA) still relies on random query selection from a purely exploitative agent, contradicting fundamental insights from preference elicitation literature (Viappiani and Boutilier, 2010). We demonstrate that the fundamental problem lies in reward modeling itself: single-agent approaches lack behavioral diversity, leading to increasingly similar trajectory pairs that provide diminishing information. In principle, if reward modeling were sufficiently accurate, explicit diversity terms might be unnecessary (Novoseller et al., 2020), but this is not the case in practice as shown in Fig. 2.

Fig.2 compares preference exploration strategies in a 2D navigation task, where the agent starts at the bottom left and the goal is at the top right. The figure shows (orange box) that trajectories from a single-agents, similar to what QPA would generate, concentrate in high-reward regions but provide limited exploration, thus hindering future queries. In contrast, our population-based method generates trajectories from multiple agents that collectively cover a wider area of the state space, including both high-reward regions and informative boundary areas. Trajectories from different population members are more distinguishable, making preference queries less ambiguous for human evaluators. Generating queries from a population of agents instead of a single neural agent might improve diversity in a few ways, because a single neural policy might have trouble representing multimodal trajectory distributions or because of the primacy bias (Nikishin et al., 2022) that might limit the representation capabilities of the agent over time. However, the population in itself is not enough to significantly improve over purely exploitative methods and we show in the figure (blue box) that without an explicit diversity bonus, the population diversity might still collapse because the neural ensemble, as used in several PbRL approaches, appears to be a poor representation of the reward posterior distribution. This qualitative observation will be further reinforced in our experiments section with quantitative results.

# 5 PB²: POPULATION-BASED PREFERENCE-BASED REINFORCEMENT LEARNING

We have discussed the importance of an explicit diversity bonus to properly explore the preference landscape. In this section, we present the different components of PB²: a **P**opulation-**B**ased approach for **P**reference-**B**ased Reinforcement Learning that effectively explores the preference landscape while maintaining alignment with human preferences. To combat diversity collapse of reward neural ensembles, we employ in PB² an information-theoretic formulation that maximizes the mutual information between policies and their state distributions. This encourages each agent to explore a distinct subset of states that can be readily identified by a discriminator, naturally generating clearly distinguishable behaviors for human evaluation. The challenge is then to to maintain effective diversity in a non-stationary context, as the interactive nature of PbRL creates a continuously evolving reward landscape, and we need to maintain diversity while remaining aligned with the current reward model, as detailed in the following sections.

## 5.1 PERFORMANCE-CONSTRAINED DIVERSITY FOR PREFERENCE EXPLORATION

PB² addresses the preference exploration challenge through a population-based approach with two key mechanisms: (1) an anchor, a reference policy that tracks achievable performance under current preferences, and (2) diverse policies that explore the preference landscape while remaining aligned with human intent.

**Performance-Constrained Diversity.** Building on the performance-constrained diversity principle introduced in SMERL (Kumar et al., 2020), we adapt this mechanism to the preference learning setting. Unlike SMERL, which requires access to an expert or optimal return value, PB² operates in settings where the reward function itself is being learned. Similarly to QPA, we learn a single reward model that plays the role of the maximum likelihood reward function. From this model, we

---

**Algorithm 1** PB²: Population-Based Preference-Based RL

---

1: Initialize anchor policy $\pi_{\text{ref}}$, diverse policies $\{\pi_i\}_{i=2}^N$, discriminator $q_\psi$, reward model $r_\phi$ *(Section 5)*
2: Perform initial unsupervised exploration to collect diverse trajectories      *(Lee et al., 2021b)*
3: **while** feedback budget not exhausted **do**
4:      Sample trajectories from all policies (anchor + diverse)      *(Section 5)*
5:      Collect human preferences on trajectory segments pairs across policies    *(Fig. 1, Section 5)*
6:      Update reward model $r_\phi$ to predict preferences      *(Section 3)*
7:      Update $\pi_{\text{ref}}$ to maximize $r_\phi(\tau)$ only      *(Section 5.1)*
8:      $R_{\text{ref}} = \mathbb{E}[R_\phi(\tau)]$ for trajectories from $\pi_{\text{ref}}$      *(Section 5.1)*
9:      **for** each diverse policy $\pi_i, i \geq 2$ **do**
10:         $R_i = \mathbb{E}[R_\phi(\tau)]$ for trajectories from $\pi_i$      *(Section 5.1)*
11:         **if** $R_i \geq \alpha \cdot R_{\text{ref}}$ **then**
12:             Update $\pi_i$ to maximize $r_\phi + \lambda \cdot \log q_\psi(i|.)$ using SAC    *(Section 5.1)*
13:         **else**
14:             Update $\pi_i$ to maximize $r_\phi$ only using SAC      *(Section 5.1)*
15:         **end if**
16:      **end for**
17:      Train $q_\psi$ to maximize $\mathbb{E}_{i \sim p(i), s \sim \pi_i}[\log q_\psi(i|s)]$      *(Section 5.1, Eq. 4)*
18: **end while**

---

generate additional reward 'samples' by mixing the maximum likelihood model with diversity bonus functions.

To balance maximization of the learned reward model and the diversity bonus, our approach maintains a reference policy (the anchor policy $\pi_{\text{ref}}$) that purely maximizes the current reward model, establishing a performance baseline, which we denote by $R_{\text{ref}} = \mathbb{E}_{\tau_{\text{ref}} \sim \pi_{\text{ref}}}[R_\phi(\tau_{\text{ref}})]$. The remaining policies are encouraged to develop distinct behaviors through a discriminator $q_\psi$ that predicts which agent generated a given state, but only when their performance is within a specified threshold of the anchor:

$$\pi_i^* \in \arg\max_{\pi_i} \mathbb{E}_{s,a \sim \pi_i}\left[ r_\phi(s,a) + \lambda \cdot 1_{[R_i \geq \alpha \cdot R_{\text{ref}}]} \cdot \log q_\psi(i|s) \right] \tag{3}$$

where $r_\phi$ is the learned reward function, $R_i = \mathbb{E}[\sum_t \gamma^t r_\phi(s_t, a_t)]$ is agent $i$'s expected return, $1_{[R_i \geq \alpha \cdot R_{\text{ref}}]}$ is an indicator function that equals 1 when the agent's expected return is at least $\alpha$ times the anchor's return, $p(i) = 1/N$ is the uniform prior over the $N$ agents in the population and $q_\psi$ is a learned discriminator that predicts which agent generated a given state.

**Adaptive Discriminator.** To maintain diversity as preferences evolve, we employ a discriminator trained to maximize the mutual information between policies and their state distributions:

$$\mathcal{L}_{\text{disc}}(\psi) = \mathbb{E}_{i \sim p(i), s \sim \pi_i}[\log q_\psi(i|s)] \tag{4}$$

Unlike approaches with fixed reward objectives, our discriminator adapts continuously to the changing landscape of behaviors. This adaptation is crucial when human preferences shift, as it encourages policies to discover new distinguishable behaviors that remain aligned with current preferences. When the reward model $r_\phi$ is updated with new preferences, we temporarily disable the diversity bonus, allowing policies to first adapt to the new reward landscape before reintroducing diversity incentives. This prevents diversity from interfering with the initial adaptation to updated preferences. Fig.2 illustrates how this mechanism results in each agent receiving distinct exploration bonuses, creating a diverse set of behaviors that collectively span the preference space. The non-stationary nature of PbRL introduces unique implementation challenges such as managing discriminator behavior during preference transitions, preventing destabilizing exploration based on outdated preferences, and ensuring population coherence when the underlying reward landscape shifts with new human feedback, challenges that do not arise in vanilla RL with fixed reward functions.

Implementation details, including network architectures, hyperparameter values, stability mechanisms, and training procedures are provided in Appendix A. We next present the experimental validation of our approach.

# 6 EXPERIMENTS

We evaluate PB² across diverse environments to demonstrate its effectiveness in preference space exploration and learning from limited human feedback.

## 6.1 EXPERIMENTAL SETUP

**Environments and Baselines.** We use two environment categories: (1) **Low-feedback environments** (2D Navigation and PointMaze) for testing exploration efficiency, and (2) **DMControl locomotion tasks** (Tassa et al., 2018) for evaluating distinguishability. We compare against four leading PbRL methods: *PEBBLE* (Lee et al., 2021b), which samples queries from previous policies; *RUNE* (Liang et al., 2022), which uses ensemble-based uncertainty for exploration; *QPA* (Hu et al., 2024), which focuses on queries from recently generated trajectories; and *RIME* (Cheng et al., 2024), which handles noisy preferences through denoising discriminators. We selected these specific methods as they represent the primary query selection strategies in PbRL literature: historical sampling, uncertainty-driven exploration, current-policy alignment, and noise handling. We also compare against the naive implementation of dueling posterior sampling TS (Novoseller et al., 2020) as a population based method for comparison.

**Evaluation methodology.** For all experiments, we use the ground truth reward (which is hidden from the learning algorithms) to evaluate performance. We report the mean and standard deviation across 5 random seeds. For simulating human evaluation inconsistency, we implement the Equal teacher from B-Pref (Lee et al., 2021a) with a similarity threshold mechanism. Specifically, for a query comparing trajectories with ground truth returns $R_1$ and $R_2$, we assign random preference labels when $|R_1 - R_2| < \epsilon \cdot \max(R_1, R_2)$, where $\epsilon$ is the similarity threshold parameter. This approach only introduces inconsistency when trajectories are genuinely similar and difficult to distinguish, better simulating human evaluation challenges. While $\epsilon = 0$ (no similarity threshold) provides a theoretical baseline, we emphasize results with $\epsilon > 0$ as more realistic, since human evaluators inevitably provide less consistent feedback when comparing similar behaviors.

## 6.2 EXPLORATION EFFICIENCY IN LOW-FEEDBACK ENVIRONMENTS

**Escaping Local Optima in Preference Landscapes** Fig.3 illustrates a failure case for single-agent methods and how PB² overcomes it. After identical initial feedback (4 queries), both algorithms learn similar reward models favoring an upper-left region. After 20 feedback instances, QPA remains trapped in this suboptimal region due to its exploitative behavior, while PB² successfully discovers a path to the high-reward region through its diverse population. The discriminator encourages different agents to explore distinct regions while maintaining performance, allowing discovery of paths that eventually lead to optimal areas. This demonstrates a key advantage of population-based methods: when facing potentially suboptimal initial queries, PB² can escape local optima through diversity, while single-agent approaches often remain trapped. The complete progression of trajectories with increasing feedback iterations is provided in appendix, showing the step-by-step evolution of each method's exploration patterns.

**Preference Exploration Improves Feedback Efficiency** Table 1 presents performance with varying amounts of feedback in navigation tasks. In 2D Navigation, PB² achieves significantly better performance than QPA with limited feedback (N=4 to N=8), with improvements of up to 50% at some feedback levels. As feedback increases to N=10, the gap narrows, suggesting that with sufficient feedback, the exploration advantage becomes less critical.

In the more complex PointMaze environment, PB² consistently outperforms QPA at most feedback levels (N=12, N=16, N=20), with advantages of 20-30% in some cases. Overall, these results confirm that maintaining a diverse population enables more efficient exploration of complex preference landscapes when feedback is scarce, a crucial advantage in applications where minimizing human interaction is essential. Note that the N=2 for 2D Navigation and N=4 for PointMaze represents per-

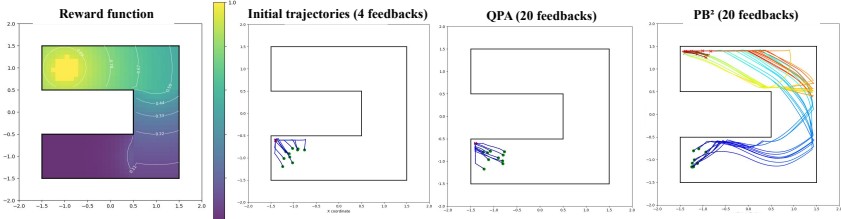

Figure 3: **Escaping local optima in preference landscapes.** After the same initial feedback following unsupervised exploration (4 queries), both algorithms learn similar reward models favoring the upper-left region. After 20 feedback instances, QPA (middle) remains trapped in this suboptimal region due to its exploitative behavior, while PB² (right) successfully discovers a path to the high-reward region (left) through its diverse population approach.

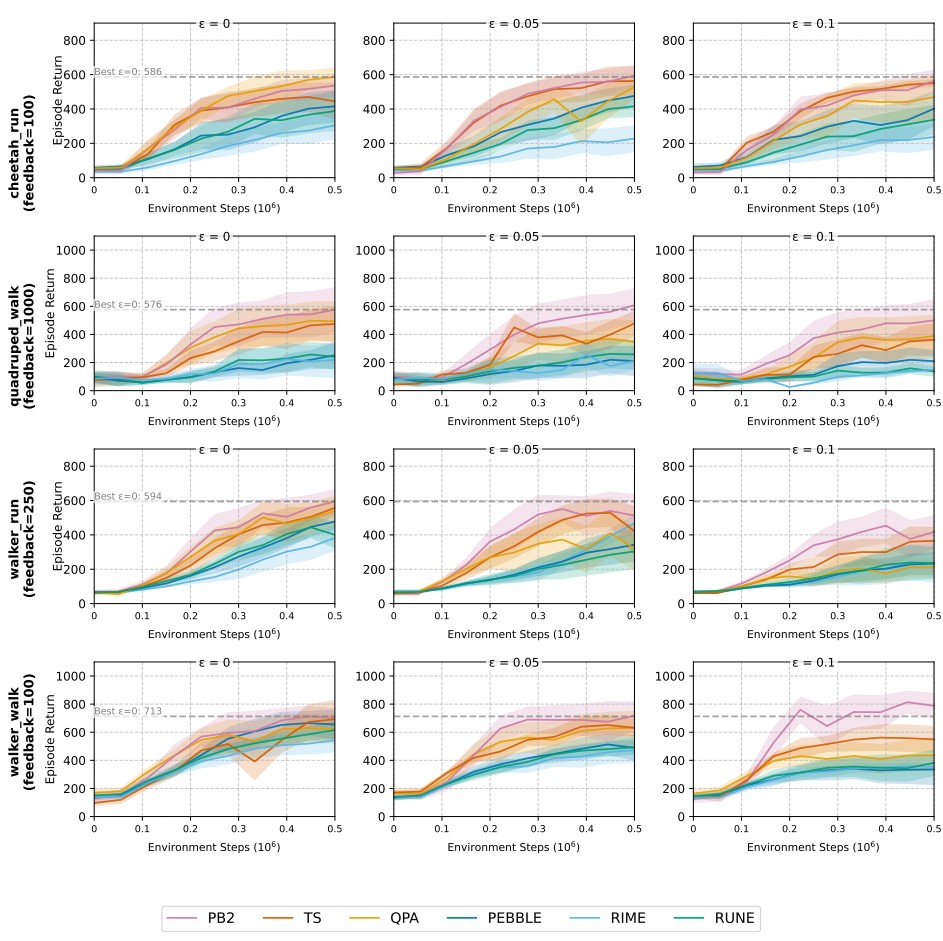

Figure 4: **Performance comparison on DMControl tasks with varying similarity thresholds.** Each column represents a different similarity threshold $\epsilon$ that determines when trajectory comparisons result in inconsistent feedback. As $\epsilon$ increases from 0 (perfect oracle) to 0.1 (significant inconsistency with similar trajectories), PB² (pink) maintains robust performance while single-agent and the naive posterior sampling methods degrade more substantially, particularly in the quadruped_walk and walker_walk tasks. This demonstrates PB²'s advantage in avoiding error prone queries.

Table 1: Performance Comparison in 2D Navigation and PointMaze environments by Feedback Amount (N)

| | Algorithm | Feedback Amount | | | |
| | | N=4 | N=6 | N=8 | N=10 |
|---|---|---|---|---|---|
| 2D Navig. | PB² | $\mathbf{-211.3 \pm 58.7}$ | $\mathbf{-141.0 \pm 48.5}$ | $\mathbf{-178.7 \pm 70.7}$ | $-126.5 \pm 18.9$ |
| | QPA | $-405.7 \pm 164.6$ | $-285.4 \pm 160.3$ | $-234.8 \pm 154.9$ | $\mathbf{-112.1 \pm 24.4}$ |
| | PEBBLE | $-323.6 \pm 66.0$ | $-156.1 \pm 41.2$ | $-196.3 \pm 95.9$ | $-164.5 \pm 71.1$ |
| | RUNE | $-359.8 \pm 78.5$ | $-204.6 \pm 109.6$ | $-251.45 \pm 81.5$ | $-147.08 \pm 62.6$ |
| | TS | $-320.5 \pm 95.1$ | $-167.9 \pm 45.6$ | $-214.3 \pm 78.8$ | $-115.9 \pm 14.0$ |
| | | N=8 | N=12 | N=16 | N=20 |
| PointMaze | PB² | $18.9 \pm 3.63$ | $\mathbf{85.0 \pm 45.3}$ | $\mathbf{132.2 \pm 28.7}$ | $\mathbf{146.1 \pm 29.3}$ |
| | QPA | $30.7 \pm 37.0$ | $63.6 \pm 55.0$ | $116.4 \pm 18.5$ | $110.6 \pm 14.2$ |
| | PEBBLE | $\mathbf{33.2 \pm 30.5}$ | $80.6 \pm 63.1$ | $91.4 \pm 60.4$ | $98.1 \pm 52.5$ |
| | RUNE | $36.3 \pm 36.3$ | $57.9 \pm 38.3$ | $65.5 \pm 38.6$ | $90.2 \pm 49.9$ |
| | TS | $30.3 \pm 57.3$ | $72.7 \pm 26.3$ | $117.5 \pm 70.6$ | $107.2 \pm 75.2$ |

formance after initial unsupervised exploration and random query selection, before the distinguishing characteristics of each algorithm have meaningful impact on the learning process.

## 6.3 Evaluating distinguishability in high-dimensional continuous control

**Robustness to Trajectory Distinguishability Challenges.** Fig.4 compares performance across DMControl tasks with three trajectory similarity thresholds ($\epsilon \in \{0, 0.05, 0.1\}$). With $\epsilon = 0$ (oracle teacher, no similarity consideration), PB² and QPA achieve comparable performance. However, as the similarity threshold increases, PB² maintains stronger performance, particularly in quadruped_walk and walker_walk. On the latter task with $\epsilon = 0.1$, PB² achieves a return of approximately 750 compared to around 400 for QPA and 350 for PEBBLE. By presenting more diverse and distinguishable trajectory segments pairs for evaluation, PB² makes it easier for humans to provide consistent, reliable feedback, offering a significant advantage in real-world settings where trajectories may be similar and difficult to differentiate. Interestingly, the diversity bonus is key to this robustness. Indeed, we added for this experiment an extra baseline that uses the same population-based setting of PB², but replaces explicit diversity enhanced reward functions with a neural ensemble, trained following standard PbRL procedures (Christiano et al., 2017; Lee et al., 2021b), that provides to each agent its own reward model. This baseline represents what would have been a more naive implementation of dueling posterior sampling (Novoseller et al., 2020), and we label it TS (for Thompson Sampling). Despite using an ensemble of reward functions and a population of agents instead of the single agent single reward function of QPA, TS ends-up performing very similarly to QPA, reinforcing our suspicions discussed in Sec. 4 that a vanilla neural ensemble poorly captures the posterior reward distribution.

## 7 Limitations and Future Work

**Scalability with Population Size.** Our implementation uses a small population (3 agents) due to practical constraints in the preference learning setting. With feedback budgets typically limited to approximately 10 evaluations per iteration, larger populations would reduce the number of learning iterations possible. Our analysis in Appendix Fig. 7 shows that 3-4 agents achieve optimal performance, with larger populations showing diminishing returns due to the human feedback bottleneck. Future work could explore adaptive population sizing strategies that balance diversity benefits against feedback constraints in different preference landscapes.

**Exploration-Exploitation Tradeoff hyperparameter.** PB² introduces a new hyperparameter, $\lambda$, that controls the balance between diversity and reward optimization. Our sensitivity analysis in Appendix Fig. 6 shows that moderate values perform well while extreme values converge to baseline performance. This robustness is ensured by our performance-constrained diversity mechanism (Equation 3), which automatically disables the diversity bonus when it hurts performance, preventing catastrophic failures and guaranteeing at least single-agent performance regardless of $\lambda$ choice. To ensure fair evaluation, we kept $\lambda$ consistent across all environments within each class (locomotion tasks and navigation tasks), rather than tuning it per environment. Nevertheless, requiring different values for each environment classes remains a limitation, as individually tuning $\lambda$ for each specific

environment could potentially yield better performance. Future work could develop adaptive methods that automatically adjust this exploration-exploitation tradeoff based on the current state of preference learning, eliminating the need for domain-specific parameter tuning, though developing robust adaptation criteria remains an open research question.

## 8 CONCLUSION

We presented PB², a population-based approach for preference-based reinforcement learning that addresses the challenge of preference space exploration. By maintaining diverse agents that collectively explore the preference landscape, PB² generates more distinguishable behaviors for human evaluation, improving reward model learning efficiency and robustness to evaluation inconsistencies. Our experimental results demonstrate three key advantages: improved feedback efficiency with limited feedback, greater robustness to labeling inconsistency, and enhanced ability to escape local optima in complex preference landscapes. These benefits make PB² particularly well-suited for real-world applications where human feedback is costly and potentially inconsistent.

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

# A  IMPLEMENTATION DETAILS

## A.1  PB² ALGORITHM

In this section, we provide the full procedure for PB², our population-based approach for preference-based reinforcement learning, in Algorithm 2.

## A.2  POPULATION MANAGEMENT

### A.2.1  POPULATION DESIGN CHOICE

Unlike methods such as DIAYN and SMERL that use a single policy network conditioned on a latent variable to learn multiple behaviors, PB² maintains separate policy networks for each agent in the population. This design choice offers key advantages: it allows for independent update of population members, enables different agents to fulfill distinct roles (exploration vs. exploitation) and provides a robust mechanism for recovery when individual agents fail. Having a dedicated anchor that focuses solely on maximizing the learned reward function creates a stable anchor for the population, serving as a reliable fallback that can replace underperforming explorer agents when necessary.

### A.2.2  REFERENCE AGENT APPROACH

One of the key innovations in PB² is our anchor agent mechanism, which provides a stable performance benchmark without requiring access to an expert policy or ground-truth reward function. This section details the complete implementation of this approach.

The anchor (indexed as agent 0 in our implementation) is trained to maximize only the current reward model without any diversity bonus:

$$\pi_0 \in \arg\max_{\pi} \mathbb{E}_{\tau \sim \pi} \left[ \sum_{t=0}^{T} \gamma^t r_\phi(s_t, a_t) \right] \tag{5}$$

The reward model $r_\phi$ is continuously updated based on human preference feedback. After each reward model update, we track the anchor's expected return under the new reward function:

$$R_{\text{ref}} = \mathbb{E}_{\tau \sim \pi_0} \left[ \sum_{t=0}^{T} \gamma^t r_\phi(s_t, a_t) \right] \tag{6}$$

In practice, this expectation is approximated by maintaining a running average of episode returns for the anchor over a window of recent episodes.

The other agents in the population (indexed 1 through $N-1$) are trained with the following objective:

$$\pi_i \in \arg\max_{\pi} \mathbb{E}_{\tau \sim \pi} \left[ \sum_{t=0}^{T} \gamma^t \left( r_\phi(s_t, a_t) + \lambda \cdot 1_{[R_\phi(\tau) \geq \alpha \cdot R_{\text{ref}}]} \cdot \log q_\psi(i|s_t) \right) \right] \tag{7}$$

Where:

- $\lambda$ is the diversity coefficient (typically 0.25 for locomotion tasks, 0.5 for navigation)
- $\alpha$ is the performance threshold (0.9 in our implementation)
- $1_{[R_\phi(\tau) \geq \alpha \cdot R_{\text{ref}}]}$ is an indicator function that equals 1 when the agent's expected return is at least $\alpha$ times the anchor return
- $q_\psi(i|s)$ is the discriminator that predicts which agent generated a given state

In practice, the discriminator $q_\psi(i|s)$ is implemented as a neural network that takes states as input and outputs a probability distribution over the $N$ agents. The diversity reward $\log q_\psi(i|s)$ is computed for each state encountered during training and added to the extrinsic reward $r_\phi(s, a)$ when the

performance constraint is satisfied. In our implementation, we track the recent performance of each agent and apply the diversity bonus only when an agent's performance exceeds the threshold relative to the anchor. This performance-constrained diversity mechanism ensures that exploration through diversity is only encouraged when it doesn't significantly compromise task performance.

### A.2.3 Policy Inheritance in the Population

After each feedback session in PB², we transfer the critic and actor networks from the anchor to all explorer agents in the population. This inheritance mechanism preserves the core task knowledge while still enabling agent specialization through the diversity bonuses. It serves multiple purposes:

It allows all agents to quickly benefit from updated reward models based on human feedback, ensuring the entire population adapts to new preference information simultaneously. It maintains a healthy balance between exploration and exploitation across the population, with the anchor agent focusing on exploitation while explorer agents develop diverse behaviors from a common foundation. Most importantly, it prevents explorer agents from drifting too far from the task objective due to diversity pressures, which could otherwise lead to behaviors that are novel but ineffective.

This periodic knowledge sharing proved particularly beneficial in complex environments where unguided exploration can be inefficient, allowing the population to maintain both performance and diversity throughout training.

### A.3 Stability Mechanisms

### A.3.1 Transition Handling Between Preference Updates

A critical challenge in our population-based approach is managing the discriminator's behavior during preference transitions. When the reward model updates, we implement several specific techniques to ensure stable and effective diversity guidance:

1. **Temporary Diversity Suspension:** Immediately after a reward model update, we temporarily disable the diversity bonus by setting the discriminator coefficient to zero for a fixed period (typically 5000 environment steps). This suspension period allows all agents to first adapt to the new reward landscape before reintroducing diversity incentives, preventing potentially counterproductive exploration based on outdated preferences.

2. **On-Policy Discriminator Training:** During transition periods, the discriminator is trained exclusively on recent experiences rather than the full replay buffer. This is implemented through our on-policy sampling mechanism that selects state samples from only the most recent trajectories for each agent.

3. **Reward Normalization and Clipping:** To prevent extreme diversity bonuses during transition periods, we implement Exponential Moving Average (EMA) normalization with a decay rate of 0.99 for the discriminator's outputs. Additionally, we clip the normalized intrinsic rewards to the range $[-2, 2]$, which prevents destabilizing spikes in the diversity bonus that could otherwise derail learning during preference transitions.

These implementation details are crucial for maintaining stable diversity during preference transitions, as they address the practical challenges of keeping a population-based system aligned with changing preference signals. By temporarily reducing diversity pressure, carefully normalizing rewards, and ensuring balanced training, we allow the discriminator to smoothly adapt to new preference landscapes without destabilizing the learning process.

## B Technical Details

### B.1 Implementation Framework

Our implementation builds upon established preference-based reinforcement learning frameworks. We extend the QPA codebase (Hu et al., 2024), which itself extends the B-Pref framework (Lee et al., 2021a) that provides core functionality for preference-based learning through the PEBBLE (Lee et al., 2021b) algorithm architecture.

To accommodate our population-based approach, we expanded this framework to manage multiple policy networks simultaneously, while incorporating the on-policy query selection mechanism introduced in QPA (Hu et al., 2024). This combination allows us to leverage on-policy query selection across a diverse population of agents, rather than limiting it to a single policy.

Each agent in our population maintains its own replay buffer for experience collection and policy optimization. We introduced additional components for our discrimination-based diversity mechanism, and policy inheritance procedures.

For comparisons against baselines, we integrated implementations of competing methods including PEBBLE (Lee et al., 2021b), RUNE (Liang et al., 2022) and QPA (Hu et al., 2024) using their official codebases, ensuring fair evaluation across all approaches. Our implementation maintains compatibility with the underlying frameworks while introducing the population management and diversity mechanisms that define PB².

## B.2 NETWORK ARCHITECTURES

For fair comparison, PB² uses the same network architectures as QPA for both the SAC algorithm and reward model, with our modifications focused exclusively on the population-based exploration mechanism.

The reward model consists of 3 hidden layers with 256 units each and LeakyReLU activations, taking state-action pairs as input and producing scalar reward predictions with tanh activation.

The policy networks follow the standard SAC architecture with 2 hidden layers of 1024 units and ReLU activations. The actor outputs action means and log standard deviations (bounded between -5 and 2), while the critic takes state-action pairs as input and outputs Q-value predictions.

Our discriminator network uses 2 hidden layers of 256 units with ReLU activations and layer normalization. The network takes states $s \in \mathbb{R}^{d_s}$ as input and outputs a probability distribution over the $N$ agents via softmax activation. Formally:

$$q_\psi(i|s) = \frac{\exp(f_\psi(s)_i)}{\sum_{j=1}^{N} \exp(f_\psi(s)_j)} \tag{8}$$

where $f_\psi : \mathbb{R}^{d_s} \to \mathbb{R}^N$ is the discriminator network. This discriminator is trained using cross-entropy loss on balanced batches of states from each policy in the population.

## B.3 HYPERPARAMETERS

We maintain consistent hyperparameters across all comparison methods for fair evaluation, with the only differences being in the population-specific parameters introduced by PB². Tables 2-6 provide a comprehensive overview of the hyperparameters used in our experiments.

The general hyperparameters (Table 2) are common across all environments, while environment-specific parameters (Table 6) highlight the varying complexity and requirements of different tasks. DMControl tasks generally required more feedback than navigation tasks due to their higher-dimensional state and action spaces, while maintaining a consistent diversity coefficient of 0.25. Navigation tasks benefited from a higher diversity coefficient (0.5) to encourage more extensive exploration of the state space.

For the discriminator-related parameters (Table 5), we performed a limited hyperparameter search and found that a learning rate of 1e-5 and hidden size of 256 worked well across environments. The reward model (Table 4) and SAC parameters (Table 3) were selected to match those used in prior work for direct comparison.

| Parameter | Description | Value |
|---|---|---|
| Discount factor ($\gamma$) | Reward discount factor | 0.99 |
| Replay buffer capacity | Maximum transitions stored | Training steps |
| Population size | Number of agents (including anchor) | 3 |
| Activation function | For all networks | tanh |
| Gradient update frequency | Updates per environment step | 1 |

Table 2: General hyperparameters used in the PB² algorithm

| Parameter | Description | Value |
|---|---|---|
| Actor learning rate | Learning rate for policy network | 5e-4 |
| Critic learning rate | Learning rate for Q-networks | 5e-4 |
| Alpha learning rate | Learning rate for temperature parameter | 1e-4 |
| Initial temperature | Initial value of entropy coefficient | 0.1 |
| Target update rate ($\tau$) | Polyak averaging coefficient | 0.005 |
| Target update frequency | Steps between target network updates | 2 |
| Actor update frequency | Steps between policy updates | 1 |

Table 3: SAC hyperparameters used in the PB² algorithm

| Parameter | Description | Value |
|---|---|---|
| Ensemble size | Number of networks in reward ensemble | 1 |
| Learning rate | Learning rate for reward model | 3e-4 |
| Number of hidden layers | Hidden layers in reward network | 3 |
| Hidden size | Units per hidden layer | 256 |

Table 4: Reward model hyperparameters

| Parameter | Description | Value |
|---|---|---|
| Batch size | States per gradient update | 256 |
| Learning rate | Learning rate for discriminator | $1 \times 10^{-5}$ |
| Hidden size | Units per hidden layer | 256 |
| Activation | Hidden layer activation | ReLU |
| Normalization | Layer normalization | Yes |
| On-policy ratio | Ratio of on-policy samples for training | 0.5 |
| Reward clipping | Range for normalized rewards | $[-2, 2]$ |
| EMA decay | Decay rate for running statistics | 0.99 |

Table 5: Discriminator hyperparameters

## C   ENVIRONMENT DETAILS

To thoroughly evaluate our population-based approach for preference-based RL, we selected environments that specifically challenge the two key aspects of our method: efficient preference space exploration and robustness to human feedback inconsistency. Our environment selection targets two critical scenarios: (1) low-feedback settings where efficient exploration is crucial, and (2) complex environments where trajectory similarity makes human evaluation difficult and error-prone.

Navigation tasks provide intuitive visualization of exploration patterns and allow us to demonstrate how our method escapes local optima in preference landscapes with limited feedback. DMControl tasks, with their high-dimensional state-action spaces, create scenarios where trajectories can appear similar to human evaluators despite having different underlying rewards, testing our method's ability to generate distinguishable queries and handle inconsistent feedback.

While previous methods often include robotic manipulation tasks from Meta-World, these environments typically require substantially more feedback (often 2,000-3,000 queries) to achieve meaningful

| Environment | Feedback Budget | Total Steps | Diversity $\lambda$ | Other Parameters |
|---|---|---|---|---|
| Cheetah_run | 100 | 500,000 | 0.25 | Segment length: 50 |
| Walker_walk | 100 | 500,000 | 0.25 | Unsupervised steps: 9,000 |
| Walker_run | 250 | 500,000 | 0.25 | Interact steps: 20,000 |
| Quadruped_walk | 1000 | 500,000 | 0.25 | Queries per iteration: 10 |
| 2D Navigation | 10 | 20,000 | 0.5 | Segment length: 20-50 |
| PointMaze | 20 | 80,000 | 0.5 | Unsupervised steps: 900-400 |
| | | | | Interact steps: 2,000-10,000 |
| | | | | Queries per iteration: 2-4 |

Table 6: Environment-specific hyperparameters

performance. Such high feedback requirements are unrealistic in practical human-in-the-loop scenarios. Nevertheless, we include one high-feedback experiment (1,000 queries) on the Quadruped_walk task to demonstrate our method's performance across the feedback spectrum.

## C.1 NAVIGATION TASKS

### C.1.1 2D NAVIGATION

The 2D navigation environment consists of a $10 \times 10$ continuous arena where the agent starts at the bottom left corner $(0, 0)$ and must navigate to a goal position at the top right corner $(10, 10)$. The state space is 2-dimensional, corresponding to the agent's $(x, y)$ position. The action space is also 2-dimensional, where actions directly change the agent's position with values in the range $[-1, 1]$. If the agent attempts to move outside the boundaries of the arena, it is projected to the closest point inside. The reward function used for ground truth evaluation (not accessible to the agent) is the negative Euclidean distance to the goal position.

For human feedback simulation, we compare trajectory segments of length 50 timesteps. The oracle provides preferences based on the total progress made toward the goal during each segment. When the similarity threshold $\epsilon$ is applied, random labels are provided when the difference in progress between segments falls below the threshold.

### C.1.2 POINTMAZE

In the PointMaze environment, a point mass agent navigates through a maze with walls to reach a designated goal location. The state space consists of the agent's position and velocity (4D). The action space is 2-dimensional, controlling the force applied in the $x$ and $y$ directions.

Instead of using the original reward function based on Euclidean distance to the goal, we replace it with a handcrafted reward function that better aligns with human preferences by guiding the agent through the maze as shown in Fig.3. This reward function provides higher values along the correct path through the maze corridors, creating a more structured reward landscape that captures the preference for following the intended route rather than attempting to move directly toward the goal (which would cause collisions with walls). This modification helps simulate realistic human preferences that incorporate domain knowledge about the maze's structure rather than simple distance metrics.

## C.2 DMCONTROL TASKS

We evaluate our method on four continuous control tasks from the DeepMind Control Suite (DM-Control) (Tassa et al., 2018): Cheetah_run, Walker_run, Walker_walk, and Quadruped_walk. These environments feature continuous state and action spaces with increasing complexity, from the 17-dimensional state space of Cheetah_run to the 78-dimensional state space of Quadruped_walk. All DMControl tasks have episode lengths of 1000 timesteps.

The ground truth rewards in these environments, which are used only for evaluation and not accessible to the learning algorithms, combine task-specific objectives (such as forward velocity above

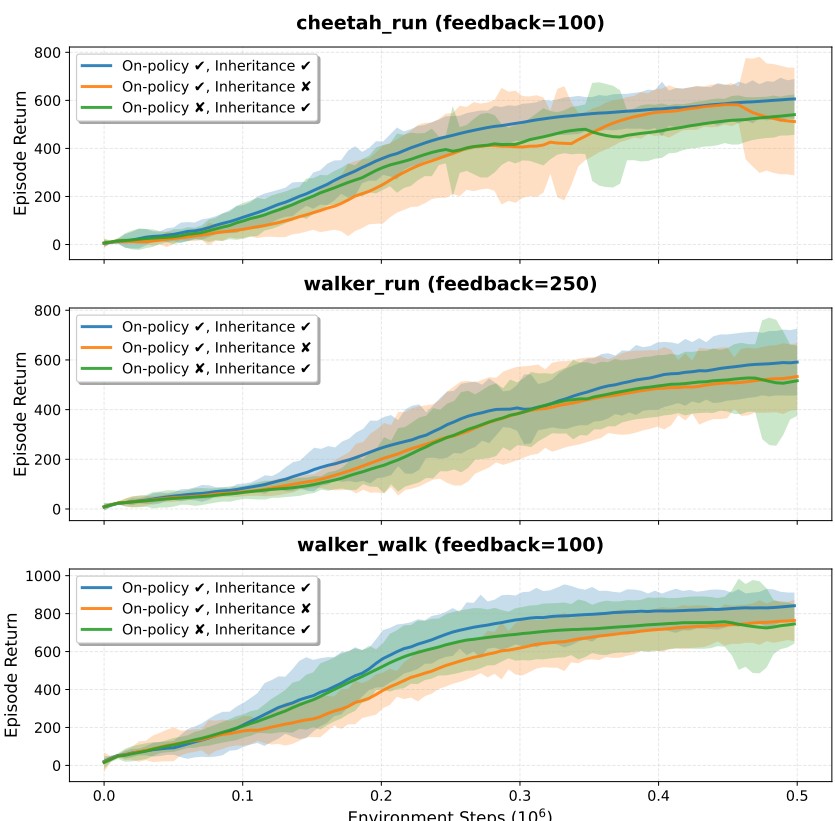

Figure 5: Ablation study on DMControl locomotion tasks showing the contribution of key components in PB². Results demonstrate that both on-policy query generation and agent inheritance mechanisms are essential for achieving optimal performance across different environments and feedback budgets.

environment-specific thresholds) with control penalties. The Walker and Quadruped environments additionally reward upright posture maintenance.

# D  ADDITIONAL EXPERIMENTAL RESULTS

## D.1  COMPONENT ABLATION STUDY

To evaluate the contribution of individual components in PB², we conducted ablation studies examining the impact of on-policy sampling and policy inheritance mechanisms. Figure 5 shows results across three DMControl environments.

The results demonstrate that both on-policy sampling and inheritance contribute positively to performance. The full PB² method (On-policy , Inheritance ) consistently achieves the highest performance across all environments. Removing policy inheritance (On-policy , Inheritance ) leads to noticeable performance degradation, particularly in the later stages of training. This confirms the importance of knowledge sharing between agents in the population. Interestingly, removing on-policy sampling while keeping inheritance (On-policy , Inheritance ) shows competitive performance in some environments, suggesting that the benefits of these components can be complementary depending on the task complexity.

## D.2  DIVERSITY PARAMETER ANALYSIS

We investigated the effect of the diversity parameter $\lambda$ on learning performance. Figure 6 shows results for the Walker_walk environment with different values of $\lambda$ ranging from 0 to 1.

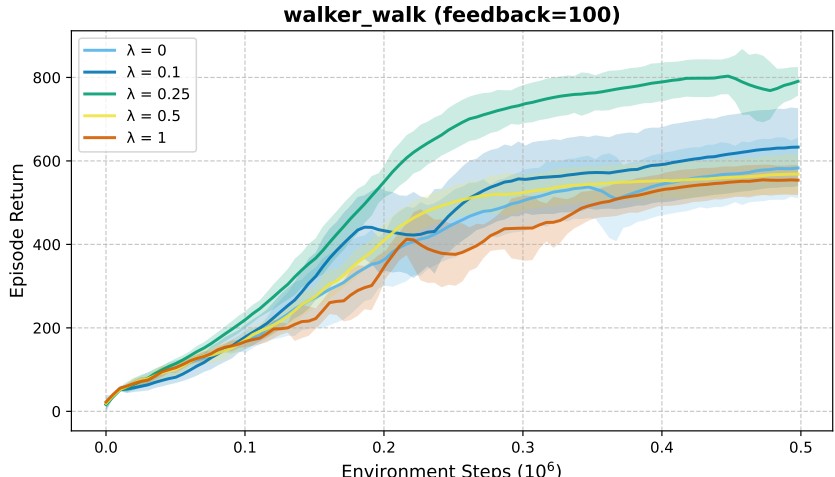

Figure 6: Sensitivity analysis of the diversity parameter $\lambda$ in the walker_walk environment with 100 feedback queries. The results show that small values of $\lambda$ (0.1-0.25) achieve the best balance between exploration and exploitation in this setup, while extreme values ($\lambda = 0$ or $\lambda = 1$) lead to suboptimal performance.

The results reveal that moderate values of $\lambda$ (0.1-0.25) achieve the best performance, with $\lambda = 0.25$ showing the strongest results. Setting $\lambda = 0$ (no diversity bonus) leads to reduced performance due to insufficient exploration and limited behavioral diversity across the population. Conversely, very high values ($\lambda = 1$) also underperform, suggesting that excessive emphasis on diversity can distract agents from optimizing the primary reward signal. The optimal range around $\lambda = 0.25$ (for the current environment) provides an effective balance between reward maximization and exploration diversity, enabling agents to discover distinct yet effective behaviors.

### D.3 POPULATION SIZE SENSITIVITY

Figure 7 examines the impact of different population sizes (2, 3, 4, 5 agents) on learning performance in the Walker_walk environment.

The results show that population sizes of 3-4 agents achieve the best performance, with diminishing returns as population size increases further. A population of size 2 shows competitive early performance but fails to maintain the same final performance level as larger populations. This is likely because with a fixed query budget of 10 per iteration, having only 2 agents results in trajectories that are too similar to each other, limiting the diversity of preference queries. Conversely, populations of size 5 do not provide significant benefits over size 4, potentially because the available queries become too sparse across agents, reducing the learning signal for each individual policy. The population size of 3, which we use in our main experiments, appears to be well-chosen based on this analysis, providing sufficient diversity while maintaining concentrated learning signals.

### D.4 ANCHOR THRESHOLD SENSITIVITY

The anchor threshold parameter $\alpha$ controls when diverse agents receive the diversity bonus by setting a performance threshold relative to the anchor agent. Figure 8 demonstrates how different values of $\alpha$ (0.5, 0.7, 0.9) affect agent behavior in a 2D navigation environment. Higher $\alpha$ values create stricter conditions for agents to receive the diversity bonus, requiring them to achieve higher performance relative to the anchor before being allowed to optimize for both the reward model and diversity bonus simultaneously. When agents meet this threshold, their trajectories aim for regions that are high in both the reward model predictions (shown in yellow in the leftmost plot) and their individual diversity bonuses. With $\alpha = 0.5$, the lenient threshold allows agents to quickly access diversity bonuses, leading to trajectories that target the intersection of high reward and high diversity regions. At $\alpha = 0.7$, agents show more selective alignment to these intersection regions. At $\alpha = 0.9$, the strict

Figure 7: Impact of population size on learning performance in the walker_walk environment with 100 feedback queries. Results indicate that a population size of 3-4 agents provides the optimal trade-off between diversity benefits and computational efficiency, with diminishing returns for larger populations.

performance requirement means agents must demonstrate strong reward model alignment before they can optimize for the combination of both rewards, resulting in trajectories that more carefully target regions where both the reward model and diversity bonus are simultaneously high. The diversity bonus visualizations (top row) show distinct high-value regions for each agent, and the trajectories demonstrate how agents navigate toward areas where these bonuses overlap with reward model predictions.

### D.5 Detailed Trajectory Evolution in Point Maze Environment

This section provides the complete trajectory progression referenced in Fig.4 of the main text, showing the step-by-step evolution of exploration patterns as feedback increases from N=4 to N=16 queries in the Point Maze environment.

Figure 9 illustrates how PB² (red box) and QPA (blue box) evolve their exploration strategies with increasing feedback. At N=4, both methods show similar initial exploration patterns after receiving identical feedback. However, as feedback increases, the population-based approach in PB² enables the three agents to explore distinct regions of the maze, attempting to find high reward regions. In contrast, QPA's single-agent approach becomes increasingly concentrated in the initially promising but suboptimal upper-left region, demonstrating the local optima problem discussed in the main text.

Crucially, while not all agents in PB² necessarily discover the optimal path simultaneously, once one agent finds a better trajectory (Agent 3, N=12), the reward model update incorporates this improved knowledge by comparing it against the previous suboptimal behaviors from other agents. This allows the shared reward model to capture the superior strategy, subsequently guiding all agents toward the newly discovered high reward region (N=16, Agents 1,2 and 3). This progression clearly shows how PB²'s diverse population prevents premature convergence and enables discovery of multiple pathways that eventually lead to finding the optimal solution, while QPA remains trapped in its initial exploration pattern.

### E Computational Resources and Reproducibility

#### E.1 Compute Resources

PB² requires approximately 3-4× more computational resources than single-agent baselines due to maintaining multiple policies simultaneously. However, since our population is small (3-4 agents), this overhead primarily translates to a slightly additional GPU memory (requirements for storing

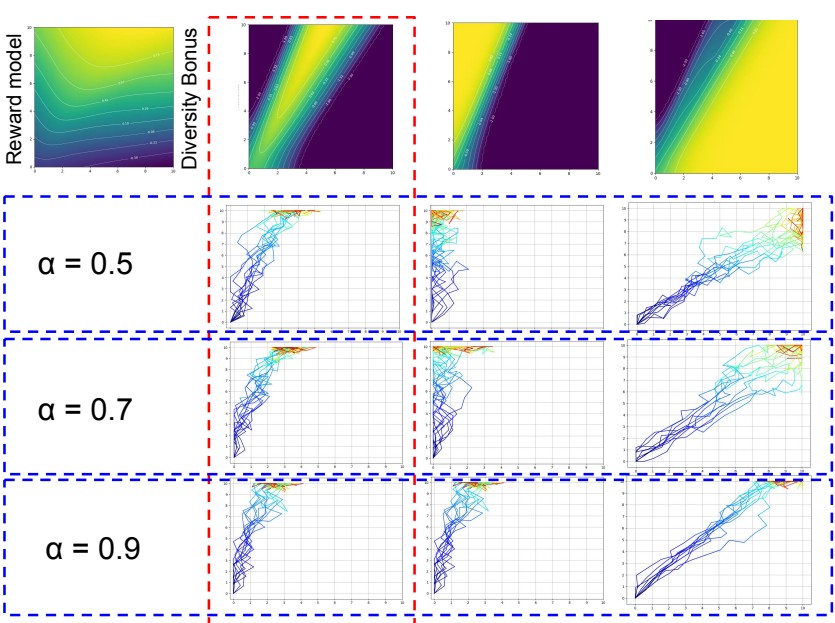

Figure 8: Anchor threshold sensitivity analysis showing the effect of $\alpha$ on multi-objective optimization in a 2D navigation task. The parameter $\alpha$ sets the performance threshold relative to the anchor agent that diverse agents must meet to receive diversity bonuses. Higher $\alpha$ values require stricter reward model alignment before agents can optimize for both reward model predictions and diversity bonuses simultaneously, leading to trajectories that target the intersection of high reward and high diversity regions. The top row shows reward model predictions and individual diversity bonuses for three agents, while trajectories demonstrate how different $\alpha$ values affect agents' ability to balance both objectives.

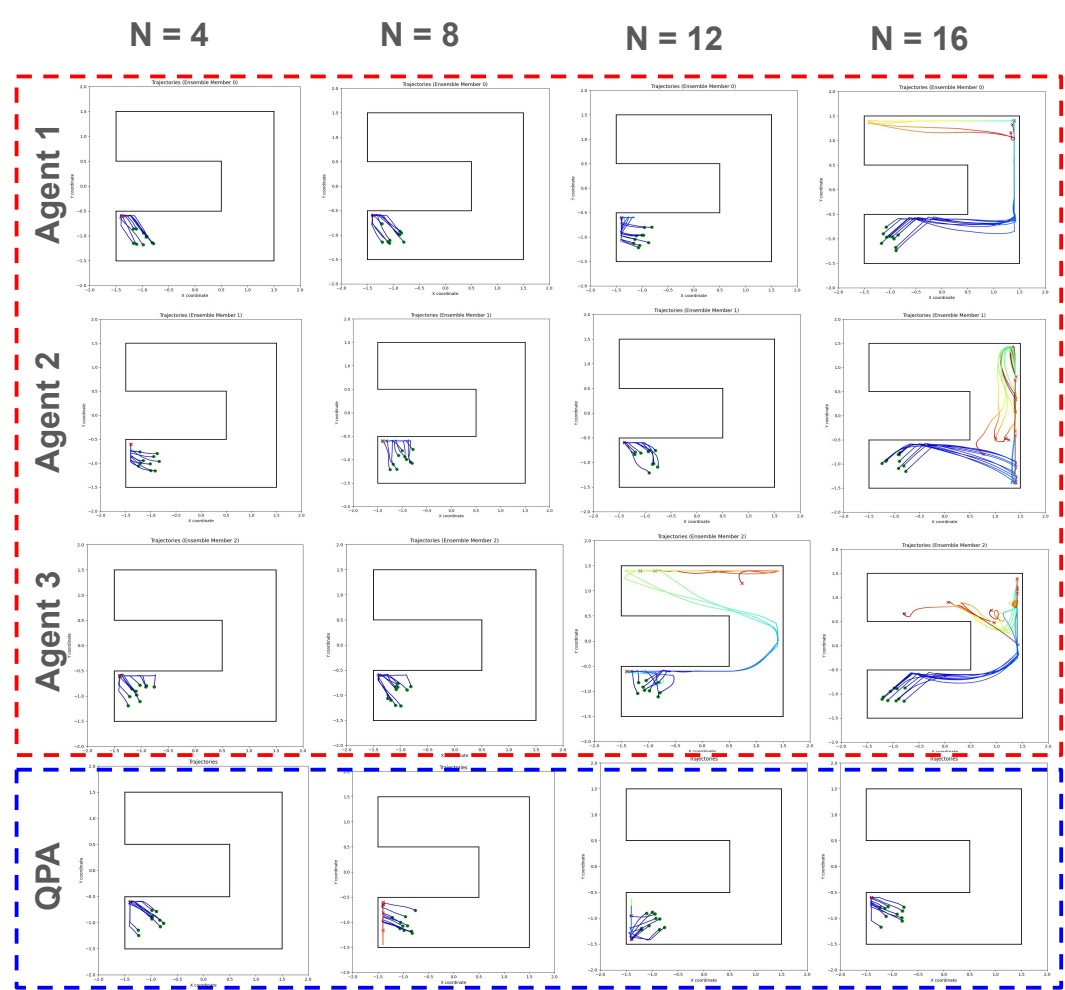

Figure 9: Complete trajectory evolution in Point Maze environment showing exploration patterns of PB² (red box, with Agent 1, 2, 3) versus QPA (blue box) as feedback increases from N=4 to N=16 queries. The progression demonstrates how PB²'s population-based approach maintains diverse exploration strategies across multiple agents, while QPA's single-agent method becomes trapped in suboptimal regions due to its exploitative nature.

multiple networks in parallel, networks being small, and some RAM for replay buffers, making the approach practically feasible.

## E.2   CODE AND DATA ACCESS

**Code Repository**   Our implementation will be made publicly available upon paper acceptance. We provide the complete codebase in the supplementary materials as a zip archive for immediate access. The codebase includes all experimental configurations, training scripts, and evaluation utilities necessary for reproduction.

---

**Algorithm 2** PB²: Population-Based Preference-Based Reinforcement Learning (Detailed)

---

1: **Initialize:** Anchor policy $\pi_{\text{ref}}$, diverse policies $\{\pi_i\}_{i=2}^N$, discriminator $q_\psi$, reward model $r_\phi$
2: Initialize replay buffers $\{B_i\}_{i=1}^N \leftarrow \emptyset$ for each policy
3: Initialize preference dataset $D \leftarrow \emptyset$
4: **for** each unsupervised pre-training step $t$ **do**
5:     **for** each policy $\pi_i$ in population **do**
6:         Collect $s_{t+1}^i$ by taking $a_t^i \sim \pi_i(a_t|s_t^i)$
7:         Compute state entropy reward $r_{\text{int}}^i \leftarrow -\log p(s_{t+1}^i)$
8:         Store transitions $B_i \leftarrow B_i \cup (s_t^i, a_t^i, s_{t+1}^i, r_{\text{int}}^i)$
9:     **end for**
10:     **for** each gradient step **do**
11:         **for** each policy $\pi_i$ in population **do**
12:             Sample minibatch $(s_j^i, a_j^i, s_{j+1}^i, r_{\text{int},j}^i)_{j=1}^B \sim B_i$
13:             Optimize policy $\pi_i$ using SAC with intrinsic reward
14:         **end for**
15:     **end for**
16: **end for**
17: **while** feedback budget not exhausted **do**
18:     **// Experience Collection Phase**
19:     **for** each environment step **do**
20:         **for** each policy $\pi_i$ in population **do**
21:             Collect $s_{t+1}^i$ by taking $a_t^i \sim \pi_i(a_t|s_t^i)$
22:             Compute reward $\hat{r}_t^i = r_\phi(s_t^i, a_t^i)$
23:             Compute discriminator reward $r_{\text{disc}}^i = \log q_\psi(i|s_t^i) - \log p(i)$
24:             Store transitions $B_i \leftarrow B_i \cup (s_t^i, a_t^i, s_{t+1}^i, \hat{r}_t^i, r_{\text{disc}}^i)$
25:         **end for**
26:     **end for**
27:     **// Feedback Collection Phase**
28:     **if** step to query preferences **then**
29:         Sample $K$ recent trajectories from each policy's replay buffer $\{B_i\}_{i=1}^N$
30:         Randomly select trajectory segments to form candidate query set $\mathcal{Q} = \{(\sigma_0, \sigma_1)\}$
31:         Collect human feedback $\{y_i\}$ for queries in $\mathcal{Q}$
32:         Store preferences $D \leftarrow D \cup \{(\sigma_0^i, \sigma_1^i, y_i)\}$
33:         **// Reward Model Update**
34:         Update reward model $r_\phi$ using dataset $D$ by minimizing loss in Equation (2)
35:         Relabel replay buffers $\{B_i\}_{i=1}^N$ using updated $r_\phi$
36:     **end if**
37:     **// Policy Optimization Phase**
38:     **// Anchor Agent Update**
39:     Calculate anchor performance $R_{\text{ref}} = \mathbb{E}_{\tau \sim \pi_{\text{ref}}}[R_\phi(\tau)]$
40:     Update anchor policy $\pi_{\text{ref}}$ to maximize $\mathbb{E}[r_\phi(s, a)]$ using SAC
41:     **// Diverse Agents Update**
42:     **for** each diverse policy $\pi_i, i \geq 2$ **do**
43:         Calculate agent performance $R_i = \mathbb{E}_{\tau \sim \pi_i}[R_\phi(\tau)]$
44:         **if** $R_i \geq \alpha \cdot R_{\text{ref}}$ **then**
45:             Update $\pi_i$ to maximize $\mathbb{E}[r_\phi(s, a) + \lambda \cdot r_{\text{disc}}^i(s)]$ using SAC
46:         **else**
47:             Update $\pi_i$ to maximize $\mathbb{E}[r_\phi(s, a)]$ using SAC (no diversity bonus)
48:         **end if**
49:     **end for**
50:     **// Discriminator Update**
51:     Collect state samples $\{s_j^i\}$ from each policy $\pi_i$
52:     Update discriminator $q_\psi$ by maximizing $\mathbb{E}[\log q_\psi(i|s_j^i)]$
53: **end while**

---

