# OpenReview forum: "PB²: Preference Space Exploration via Population-Based Methods in Preference-Based Reinforcement Learning"
_ICLR.cc/2026/Conference — Submitted to ICLR 2026_

### Official Review · Reviewer_adkb · 2025-10-28

**Soundness:** 2
**Presentation:** 3
**Contribution:** 2
**Rating:** 4
**Confidence:** 3

**Summary:**

This paper introduces PB², a population-based preference-based reinforcement learning (PbRL) framework designed to enhance exploration in the preference space. By training a diverse population of policies guided by a performance-constrained diversity bonus, PB² enables the generation of more informative and distinguishable preference queries, leading to better reward model learning and improved robustness against noisy human feedback. Experimental results demonstrate superiority over several state-of-the-art PbRL methods across various benchmarks.

**Strengths:**

1. The manuscript is clearly written and well-structured, the idea is intuitive.

2. The paper provides an insightful analysis of exploration deficiencies in existing PbRL methods, particularly highlighting how ensemble-based and single-policy approaches fail to capture sufficient behavioral diversity.

**Weaknesses:**

1. The paper lacks a formal theoretical analysis to support the claim that the proposed method effectively mitigates exploration challenges or addresses uncertainty, limiting the depth of its empirical contributions.

2. The proposed approach introduces substantial computational overhead due to the maintenance and training of multiple policy networks and a discriminator, which may hinder scalability to more complex or real-time environments.

3. The method depends on several critical hyperparameters (e.g., the diversity coefficient $\lambda$ and performance threshold $\alpha$) that are not adaptively tuned. This reliance may require domain-specific calibration, potentially limiting the method’s generalizability and ease of deployment in practice.

4. The set of baseline comparisons, while representative, could be further expanded to include more recent or diverse PbRL methods, including RIME [1], direct alignment methods like CPL [2].

[1] Cheng J, Xiong G, Dai X, et al. Rime: Robust preference-based reinforcement learning with noisy preferences. ICML 2024.

[2] Hejna J, Rafailov R, Sikchi H, et al. Contrastive preference learning: learning from human feedback without rl. ICLR 2024.

**Questions:**

see above

---

> ### Author Response · Authors · 2025-11-20
>
> We thank the reviewer for their evaluation and recognition that our manuscript is clearly written, well-structured, and provides insightful analysis of exploration deficiencies in existing PbRL methods.
>
> We address the main concerns below, followed by responses to the questions.
>
> **Weakness 1: Lack of formal theoretical analysis**
>
> We acknowledge the absence of formal theoretical guarantees. However, our empirical contributions demonstrate clear practical benefits: improved robustness to human feedback inconsistency (Figure 4), enhanced feedback efficiency in low-feedback settings (Table 1), and ability to escape local optima (Figure 3). The consistent improvements across diverse environments and conditions provide strong evidence for our approach's effectiveness. Our contribution addresses a practical problem in PbRL, the lack of behavioral diversity during query generation, through a population-based solution with comprehensive experimental validation. The distinguishable queries generated by PB² make human evaluation easier, as evaluators face clearer preference choices rather than ambiguous comparisons between similar trajectories.
>
> **Weakness 2: Computational overhead**
>
> While PB² maintains multiple policies simultaneously, the computational overhead is not prohibitive. Training can be parallelized across agents. Since we use small populations (3-4 agents), the primary overhead is memory for storing multiple networks and replay buffers, which fits within consumer GPU capabilities. Our analysis in Appendix Figure 7 shows that 3-4 agents achieve optimal performance, with larger populations showing diminishing returns due to the human feedback bottleneck.
>
> The primary bottleneck in preference-based RL is human feedback rather than computational resources. The memory overhead is justified by reduced human feedback requirements through more informative queries, improved robustness to evaluation inconsistency reducing wasted feedback, and better exploration preventing the need for corrective feedback after convergence to local optima. Since human time is typically more valuable than computational resources in preference-based settings, this tradeoff is favorable for practical applications.
>
> Regarding scalability to complex environments, if it's complexity with respect to exploration, current methods already fail in relatively simpler explorations tasks, as demonstrated in our experiments. PB² provides a better option in these scenarios. For real-time environments, we think that the challenge stems primarily from the preference-based RL setup itself rather than our specific method.
>
> **Weakness 3: Hyperparameter sensitivity**
>
> While we acknowledge λ and α require setting, our approach demonstrates reasonable robustness. We used consistent λ values within each environment class (λ=0.25 for DMControl, λ=0.5 for navigation) without per-environment tuning, demonstrating that domain-specific calibration is not required for functionality. Our sensitivity analysis (Appendix Figure 6) shows moderate values perform well while extreme values converge to baseline performance.
>
> Crucially, our performance-constrained diversity mechanism (Equation 3) provides a safety net: when diversity hurts performance, the bonus is disabled and all agents achieve at least single-agent performance. This ensures functionality regardless of λ choice. Hyperparameters do not hinder performance compared to other methods but can only improve it. Calibration is optional for further improvement, not required for basic functionality, which is why we kept the same parameters across all environments within each class.
>
> Regarding adaptive tuning, finding an objective function to automatically tune λ is challenging as it requires designing metrics that balance exploration and exploitation appropriately across different preference landscapes. While adaptive methods would eliminate manual parameter adjustment, developing robust adaptation criteria remains an open research question.

---

> ### Author Response · Authors · 2025-11-20
>
> **Weakness 4: Limited baseline comparisons**
>
> We thank you for the suggestion regarding RIME and CPL. We added RIME as a baseline since it addresses noisy preferences, providing a relevant comparison point. However, we did not include CPL as the setup differs fundamentally. CPL focuses on offline learning from static preference datasets, while our work and the selected baselines operate in the online setup without offline data.
>
> RIME and our method represent fundamentally different approaches to handling preference inconsistency. RIME detects and filters noisy labels through a denoising discriminator, while we prevent noise by generating distinguishable queries that are inherently easier for humans to evaluate consistently. Additionally, our noise model differs from RIME's setup. RIME assumes randomly corrupted labels that can be detected and filtered. In contrast, our noise arises naturally when trajectories are genuinely hard to distinguish, making the labels inherently ambiguous rather than detectably incorrect. This better simulates realistic human evaluation challenges where inconsistency stems from query difficulty rather than random annotation errors.
>
> ---
>
> Thank you for your feedback and efforts in reviewing our work. We look forward to hearing from you and providing any other clarifications during the discussion period.

---

### Official Review · Reviewer_yKcr · 2025-10-29

**Soundness:** 2
**Presentation:** 3
**Contribution:** 2
**Rating:** 4
**Confidence:** 4

**Summary:**

PB2 attempts to solve the core problem of preference space exploration. The authors point out that existing PbRL methods often suffer from premature policy convergence to suboptimal local optima. PB2 (Population-Based Preference-Based RL) is a population-based method. Its core idea is to explicitly maintain a population of agents with diverse behaviors, compelling the population to generate a series of high-return and behaviorally distinguishable trajectories to achieve efficient exploration.

**Strengths:**

• The paper accurately identifies a critical flaw in existing PbRL methods: query similarity. This is a very significant problem in real-world applications.
• The introduction of a similarity threshold, $\epsilon$, to simulate human inconsistency when facing similar trajectories is highly practical. The experiments demonstrate that PB2's robustness in high-noise (high $\epsilon$) environments far exceeds that of baseline methods.
• The experiments in the maze environment are very intuitive, showing how PB2 leverages its population diversity to successfully explore and find the globally optimal path.

**Weaknesses:**

• Compared to single-agent methods like QPA, PB2's implementation complexity and the number of hyperparameters are significantly increased. It requires maintaining N policy networks, N corresponding Q-value networks, and an additional discriminator network $q_{\psi}$. This leads to higher computational and memory overhead.
• In the grand scheme, PB2 can be seen as a diversity-based exploration strategy. I am convinced that it performs better in toy examples. However, in the DMC experiments, the variance and mean of most methods have severe overlap, making it difficult to discern a clear performance advantage for PB2.
• Appendix A.3 mentions several "tricks" to "ensure stable and effective diversity guidance." However, traditional methods like PEBBLE and QPA already perform stably without much tuning, suggesting that this added complex design exacerbates training instability.

**Questions:**

1. I am curious, compared to PEBBLE and RUNE, how would PB2 perform if it were used only as an auxiliary exploration strategy for them?
2. Does the PB2 mechanism allow me to drastically improve exploration efficiency and final performance by simply increasing the population size (e.g., to N=20)? Or would the discriminator's training become the new bottleneck at that point? Furthermore, in environments like DMC, is exploration itself truly the bottleneck?

---

> ### Author Response · Authors · 2025-11-20
>
> We thank the reviewer for their evaluation and their recognition that we accurately identify a critical flaw in existing PbRL methods, that our similarity threshold mechanism is highly practical, and that our maze experiments intuitively demonstrate how PB² leverages population diversity.
>
> We address the main concerns below, followed by responses to the questions.
>
> **Weakness 1: Implementation complexity and computational overhead**
>
> While PB² maintains multiple policies simultaneously, the computational overhead is not prohibitive. Training can be parallelized across agents. Since we use small populations (3 to 4 agents), the primary overhead is memory for storing multiple networks and replay buffers, which still fits on consumer GPUs. The primary bottleneck in preference based RL is human feedback rather than computational resources. Any additional memory cost is justified by improved robustness to human inconsistency and reduced feedback requirements. Our analysis in Appendix Figure 7 shows that 3 to 4 agents achieve optimal performance, with larger populations showing diminishing returns due to the human feedback bottleneck.
>
> Regarding hyperparameters, PB² introduces λ (diversity coefficient) and α (performance threshold), but we used consistent values within each environment class (λ = 0.25 for DMControl, λ = 0.5 for navigation) without per environment tuning. Our performance constrained diversity mechanism (Equation 3) provides robustness. In the worst case where diversity hurts performance, the bonus is disabled and all agents achieve at least single agent performance. The discriminator uses standard network architecture and training procedures.
>
> **Weakness 2: Statistical significance in DMControl experiments**
>
> We thank you for this remark and agree that some plots show overlapping confidence intervals. Based on your suggestion, we updated our results to report 10 seeds instead of 5 with 95% confidence intervals. We acknowledge that performance improvements may not always be significant at ε = 0 (perfect oracle teacher), which represents a less scenario. As you mentioned, preference space exploration is not the primary bottleneck in DMControl tasks with straightforward objectives like moving forward and staying upright. However, when we increase ε to simulate realistic human inconsistency, PB²'s advantage becomes pronounced, as shown in Figure 4, particularly in quadruped_walk and walker_walk where the gap widens substantially with ε = 0.05 and ε = 0.1. The consistent direction of improvement across tasks and increasing performance gaps with realistic noise levels suggest systematic benefits from our approach, particularly in practical scenarios where human evaluators face distinguishability challenges.
>
> **Weakness 3: Stability mechanisms as "tricks"**
>
> We appreciate this observation but would like to clarify that these mechanisms are not ad hoc fixes. The mechanisms address fundamental challenges that arise specifically from maintaining diversity in a non stationary preference learning setting. Temporary diversity suspension after reward updates prevents conflicting objectives during transitions. On policy discriminator training maintains stability during preference changes. Performance constrained diversity ensures agents never perform worse than single agent methods, maintaining alignment with human preferences.
>
> These are principled design choices for managing discriminator behavior during preference transitions, a challenge that does not arise in vanilla RL with fixed rewards or in single agent PbRL methods.
>
> **Question 1: Using PB² as auxiliary exploration for PEBBLE/RUNE**
>
> If we understand correctly, you are wondering whether using a diverse population of agents and comparing trajectories across them on top of PEBBLE or RUNE would be beneficial. We believe it should perform at least as well as current methods, if not better. PEBBLE uses random queries from the replay buffer. Comparing across agents would at minimum yield similar results with likely improvements. Similarly, RUNE uses ensemble variance as a reward bonus for single agent exploration, which could potentially benefit from population diversity.
>
> However, both PEBBLE's outdated trajectory sampling and RUNE's uncertainty driven approach have fundamental limitations beyond exploration. They generate increasingly similar queries over time regardless of selection strategy. Our contribution specifically targets the query generation process itself through population diversity, which is orthogonal to but could potentially complement these selection methods. In principle, our population based diversity mechanism could be combined with different query selection strategies. We would be open to exploring this integration in future work.

---

> ### Author Response · Authors · 2025-11-20
>
> **Question 2: Scaling population size**
>
> Our analysis in Appendix Figure 7 shows that 3 to 4 agents achieve optimal performance, with size 5 showing diminishing returns. This is directly linked to the human feedback bottleneck. With limited feedback budgets (e.g., 10 queries per iteration), larger populations spread feedback too thinly across agents, while smaller populations lack sufficient diversity. Drastically increasing population size (N=20) would likely not improve performance proportionally due to this feedback constraint, not discriminator training limitations.
>
> Regarding exploration in DMControl, we clarify the distinction between preference space exploration and state space exploration, though they are related. The diverse population aims to explore different regions of the state space to generate more informative queries, which improves preference space exploration by asking more relevant questions.
>
> SAC manages exploration and exploitation of the reward function by exploring different actions to achieve better returns. Our population manages exploration and exploitation of the preference space by balancing relevant questions to the current reward model (exploiting learned preferences) with diversity in queries (exploring new preference dimensions).
>
> We agree that preference space exploration may not be the primary bottleneck in DMControl tasks with straightforward objectives like moving forward and staying upright. However, our experiments with varying similarity thresholds (Figure 4) demonstrate that distinguishability remains critical in realistic scenarios. When ε = 0 (perfect oracle), performance gaps are smaller, but as ε increases (realistic human inconsistency), PB²'s advantage becomes pronounced. This confirms that the bottleneck is preference space exploration through diverse, distinguishable query generation, which becomes especially important when human evaluators face genuine distinguishability challenges.
>
> ---
>
> Thank you for your feedback and efforts in reviewing our work. We look forward to hearing from you and providing any other clarifications during the discussion period.

---

> > ### Comment · Reviewer_yKcr · 2025-11-27
> >
> > Thank you for your response. Most of my concerns have been addressed, so I will raise my score to 6. Good luck!

---

### Official Review · Reviewer_wJQR · 2025-10-30

**Soundness:** 3
**Presentation:** 2
**Contribution:** 2
**Rating:** 2
**Confidence:** 3

**Summary:**

PB² proposes a population-based method for PbRL to address insufficient exploration of the preference space and query ambiguity. The approach trains an anchor policy to exploit the current learned reward and multiple diverse policies encouraged via a discriminator-based mutual-information bonus under a performance constraint, yielding more distinguishable trajectories for preference queries. Experiments across navigation and DMControl tasks show improved feedback efficiency, greater robustness to noisy or inconsistent human labels, and the ability to escape local optima compared to single-agent baselines and a naïve posterior-sampling ensemble.

**Strengths:**

Maintaining behavioral diversity during the query stage can markedly increase the discriminability and stability of human comparisons, thus learning rewards more effectively and reducing feedback needs. The central problem addressed by this paper is indeed important.

The discriminator’s mutual-information objective directly optimizes for “distinguishable behaviors”, which aligns with the goal of making human comparisons easier and provides a principled mechanism for boosting information content and robustness. The design is reasonable and clear.

**Weaknesses:**

1. As acknowledged by the authors, the computationalcost of the design is significant, which may limit practical usability.
2. Other relevant approaches deserve discussion. For example, PPE [1] manages data/distribution-side coverage and evaluation reliability; and PPE appears to report stronger results than the present method. The authors should provide fuller theoretical analysis and empirical comparisons. The SENIOR paper [2] is also highly relevant.
3. The mathematical presentation lacks clarity and rigor in several places, with notable inconsistencies that should be carefully checked, including but not limited to:
   a) **Algorithm 1 vs. main equations**: Algorithm 1 Line 12 uses  $ r_\phi(\tau) + \lambda \cdot q_\psi(i), $   whereas the main method and Eqs. (3)/(4) use the **log-probability** conditioned on trajectories/states, e.g.,   $ \log q_\psi(i \mid \tau). $
   b) **Granularity of \(q_\psi\) inputs**: Algorithm 1 does not specify whether \(q_\psi\) takes entire trajectories \(\tau\) or single states \(s\); Algorithm 2 uses a state-based information-gain form,   $ \log q_\psi(i \mid s) - \log p(i). $   The paper should unify the convention in the main text and annotate Eq. (4) accordingly.
4. There are typos and minor writing issues: “one of the main claim” → “one of the main claims”; “We also the naive implementation …” (incomplete sentence); “collect diverse trajectory” → “collect diverse trajectories.” These should be carefully corrected.
5. Stronger Bayesian query selection/uncertainty-modeling baselines could and should be compared against the proposed method.


The method’s performance advantages appear limited relative to its higher computational cost, which raises concerns about practical utility. The paper would benefit from more comprehensive validation to demonstrate effectiveness. In parallel, the manuscript requires careful, rigorous proofreading to resolve notational inconsistencies and typographical issues.


[1] Zhu, Y., ... . (2024). Optimizing reward models with proximal policy exploration in preference-based reinforcement learning. In NeurIPS 2024 Workshop on Behavioral Machine Learning.

[2] Ni, H., ... . (2025). SENIOR: Efficient query selection and preference-guided exploration in preference-based reinforcement learning. arXiv preprint arXiv:2506.14648.

**Questions:**

See above.

---

> ### Author Response · Authors · 2025-11-20
>
> We thank the reviewer for their evaluation and for recognizing that our central problem is important and our design is reasonable and clear.
>
> We address the main concerns below, followed by responses to the questions.
>
> **Weakness 1: Computational cost**
>
> PB² maintains multiple policies simultaneously, but training can be parallelized across agents, making the primary overhead memory-based rather than computational time. Since our population is small (3-4 agents), this memory overhead is manageable in practice. More importantly, the primary bottleneck in preference-based RL is human feedback rather than computational resources. Since human feedback is the most costly resource, any additional computational requirements are well justified by the improved robustness to human inconsistency and reduced feedback needs that PB² provides.
>
> **Weakness 2: Missing comparisons**
>
> We appreciate the pointer to PPE and SENIOR. At the time of writing, we were not familiar with these works and thank the reviewer for bringing them to our attention. We have reviewed them and believe our work remains distinct. Both PPE and SENIOR are conceptually similar to RUNE, using different exploration bonuses to guide single-agent exploration and query selection. However, the robustness/distinguishability challenge remains since a single agent optimizing a single objective finds it difficult to generate the behavioral diversity needed for truly distinguishable queries across different reward model hypotheses. Since exploiting current preferences and exploring new ones represent separate objectives, it can be difficult to achieve them effectively with a single policy.
>
> Our population-based approach addresses this fundamental limitation by separating these objectives at a higher level, simultaneously maintaining multiple distinct behavioral strategies that span different regions of the preference landscape, targeting preference exploration through population diversity rather than single-agent exploration bonuses. Additionally, we evaluate on a robustness testing framework through similarity threshold testing (Figure 4), which explicitly measures performance degradation under realistic human evaluation inconsistency, demonstrating the practical benefits of our approach beyond performance with perfect oracle feedback.
>
> Unfortunately, code for PPE and SENIOR is not currently available, preventing direct comparison at this time. However, following another reviewer's recommendation, we have added RIME [1] as an additional baseline. RIME addresses noisy feedback through a different approach by learning to detect and handle mislabeling errors whereas we prevent errors at the source by generating more distinguishable queries that are easier for humans to evaluate consistently. We will include discussion of PPE and SENIOR in the related work and aim to provide empirical comparisons in future work when implementations become available.
>
> [1] Cheng J, Xiong G, Dai X, et al. Rime: Robust preference-based reinforcement learning with noisy preferences. ICML 2024.
>
> **Weaknesses 3 and 4: Mathematical presentation, typos**
>
> We appreciate the careful review of our notation, which has helped improve the overall clarity of the paper. We have addressed these minor inconsistencies in the revised manuscript:
>
> a) We have unified the discriminator notation throughout the paper. We have updated Equations (3) and (4) and Algorithm 1 to consistently reflect this state-based formulation.
> b) We have corrected the minor typos and grammatical issues mentioned.
>
> We believe these revisions address your concerns regarding mathematical presentation.

---

> ### Author Response · Authors · 2025-11-20
>
> **Weakness 5: Bayesian query selection baselines**
>
> While Bayesian neural networks (BNNs) [2][3] and ensembles [4] are the two most popular approaches for uncertainty quantification, ensembles typically outperform BNNs despite being considered more ad-hoc [5]. RUNE uses reward ensembles for uncertainty modeling both as an exploration bonus and as a criterion for query selection, providing a strong baseline for uncertainty-based methods. We also include TS (Thompson Sampling), a naive implementation of dueling posterior sampling, as shown in Figure 4 and Table 1.
>
> Importantly, prior work has observed that uncertainty-based query selection often provides limited benefits to policy learning, sometimes performing similarly to random query selection [6][7]
>
> [2] Max Welling and Yee W Teh. Bayesian learning via stochastic gradient langevin dynamics. ICML 2011.
>
> [3] José Miguel Hernández-Lobato and Ryan Adams. Probabilistic backpropagation for scalable learning of
> Bayesian neural networks. ICML 2015.
>
> [4] Balaji Lakshminarayanan, Alexander Pritzel, and Charles Blundell. Simple and scalable predictive uncer-
> tainty estimation using deep ensembles. Neurips 2017
>
> [5] Moloud Abdar et al. A review of uncertainty quantification in deep learning: Techniques, applications and
> challenges. Information Fusion 2021.
>
> [6] Kimin Lee, Laura M Smith, and Pieter Abbeel. Pebble: Feedback-efficient interactive reinforcement
> learning via relabeling experience and unsupervised pre-training. ICML 2021
>
> [7] Borja Ibarz, Jan Leike, Tobias Pohlen, Geoffrey Irving, Shane Legg, and Dario Amodei. Reward
> learning from human preferences and demonstrations in atari. Neurips 2018
>
> **Regarding performance advantages and computational cost**
>
> We respectfully believe the performance advantages are more substantial than suggested. Our experiments demonstrate clear benefits in realistic scenarios:
>
> - **Robustness to human inconsistency**: Figure 4 shows PB² maintains performance under trajectory similarity (ε=0.1) while baselines degrade significantly.
> - **Feedback efficiency**: Table 1 demonstrates 30-50% improvements with limited feedback in navigation tasks.
> - **Local optima escape**: Figure 3 qualitatively shows PB² discovers optimal solutions where single agent methods remains trapped
>
> Previous methods typically evaluate only with perfect oracle feedback, whereas our framework demonstrates practical benefits under more realistic conditions where human evaluators provide inconsistent labels on similar trajectories. This improved robustness through distinguishable queries and enhanced preference space exploration directly addresses critical real-world challenges. As discussed in our response to Weakness 1, the computational overhead is primarily memory-based and manageable given our small population size, while the bottleneck remains human feedback rather than compute.
>
> ---
>
> Thank you for your feedback and efforts in reviewing our work. We look forward to hearing from you and providing any other clarifications during the discussion period.

---

> > ### Comment · Reviewer_wJQR · 2025-11-25
> >
> > I thank the authors for the additional explanations and clarifications provided in the rebuttal.
> >
> > The new discussion of PPE, SENIOR, and RIME is helpful, although I am still unsure whether these are the only strongly related recent works; in my view, a more systematic and comprehensive treatment of related work is important for the final version.
> >
> > Across the reviews there also seems to be a shared concern about computational cost and implementation complexity, which may limit the practical usability of the method: the paper feels like a substantial engineering step on a very relevant real-world problem, but with non-trivial resource and usability costs attached.
> >
> > Taking the rebuttal and the other reviews into account, I am inclined to increase my score to 4.

---

> > > ### Author Response · Authors · 2025-11-25
> > >
> > > Thank you for increasing your score and for your constructive feedback throughout the review process, which has helped improve the clarity of our work. We greatly appreciate your thoughtful consideration of our responses.
> > >
> > > **Regarding related work**: We have updated the main paper to include discussion of PPE, SENIOR, and RIME, discussing how they relate to our approach. We will incorporate any additional closely related methods we find before the final version.
> > >
> > > **Regarding computational cost and usability**: We acknowledge that population-based methods introduce additional complexity compared to single-agent approaches. That said, the overhead is primarily memory-based with our small population, which remains manageable even on consumer GPUs. Our findings demonstrate that generating diverse behaviors for distinguishable queries can be effectively achieved through multiple distinct policies. This design choice achieves improvements in both robustness to inconsistent feedback and query efficiency, providing key practical benefits for real-world PbRL applications where human feedback is the limiting factor.
> > >
> > > Thank you again for the follow-up and for recognizing the overall strength of our work. We look forward to further discussion and welcome any additional questions or clarifications during the discussion period.

---

### Official Review · Reviewer_UaLC · 2025-11-03

**Soundness:** 3
**Presentation:** 3
**Contribution:** 2
**Rating:** 6
**Confidence:** 3

**Summary:**

This paper introduces PB², a novel population-based framework for preference-based reinforcement learning (PbRL) aimed at addressing the lack of behavioral diversity during user feedback collection. Traditional single-policy PbRL methods often converge to local minima in the preference space, limiting exploration and leading to suboptimal alignment with human preferences.

PB² tackles this by training multiple distinct policies simultaneously, each encouraged to explore different behavioral modes through an explicit diversity bonus. These diverse policies generate varied trajectories, which are then compared using human preference feedback to train a shared reward model. A discriminator module maintains population diversity while ensuring that learned behaviors remain consistent with user preferences.

Experimental results across DMControl locomotion and navigation tasks show that PB² produces more diverse and distinguishable behaviors, improves reward learning efficiency, and remains robust under noisy or inconsistent feedback. Additionally, the paper reveals that neural ensemble models fail to capture preference uncertainty effectively, offering little improvement over deterministic baselines.

**Strengths:**

1. The idea of separating a reference policy and a diverse policy to serve as different purposes is insightful and reasonably novel in preference-based RL literature.
2. The use of discriminator to differentiate between anchor and diverse incorporates the idea of adversarial learning into preference-based RL algorithms.
3. Experimental results demonstrate performance gains in both sample and feedback efficiency in different types of teachers, which are more aligned with real world scenarios.

**Weaknesses:**

1. Although the use of discriminator is different from previous work that only focuses on single-agent preference-based RL algorithms, are there implementation and experimental challenges as a result?

**Questions:**

1. Will the policy used for evaluation the same as the reference policy $\pi_{\text{ref}}$? Is it possible to somehow combine reference policy and diverse policy to see what are resulting behaviors, since exploration may also benefit a task-specific policy?
2. It seems that choice of $\alpha$ is based on heuristic, i.e. ideally the diversity-based exploration should contain meaningful behavior at least some portion of anchor policy. How sensitive is performance with different values of $\alpha$?
3. Is the discriminator trained to maximize mutual information between all trajectories collected by the diverse policy? Or to distinguish between the anchor policy and the remaining from diverse policy?

---

> ### Author Response · Authors · 2025-11-20
>
> We thank the reviewer for their evaluation and recognition of our approach's novelty and experimental validation under realistic feedback conditions.
>
> We address the main concerns below, followed by responses to the questions.
>
> **W1: Implementation and experimental challenges with discriminator**
>
> Yes, there are several challenges that arise from using a discriminator in the non-stationary PbRL setting:
>
> - **Discriminator behavior during preference transitions**: When the reward model updates based on new human feedback, the reward landscape shifts and the discriminator can become unstable as it tries to maintain diversity incentives based on outdated preference information. We address this through temporary diversity suspension for a fixed period after reward model updates, allowing agents to first adapt to new preferences before reintroducing diversity incentives (Appendix A.3.1).
>
> - **Overfitting to outdated behaviors**: As preferences evolve, the discriminator can overfit to past behaviors that are no longer relevant. We prevent this through on-policy discriminator training where we sample from recent experiences rather than the full replay buffer.
>
> - **Extreme diversity bonuses**: During transition periods, diversity bonuses can spike and destabilize learning. We mitigate this through exponential moving average normalization and reward clipping.
>
> - **Risk of diversity pushing agents away from task objectives**: Diversity exploration might conflict with task performance or push agents away from high-reward regions aligned with human preferences. Our performance-constrained diversity mechanism (Equation 3) ensures agents maintain diversity only when they stay within high-performing regions near the task objective. The diversity bonus is automatically disabled when performance drops which guarantees at least single-agent performance in the worst case.
>
> **Q1: Policy evaluation and combining policies**
>
> Yes, we use the reference (anchor) policy for evaluation, as it focuses solely on maximizing the learned reward model. The diverse policies primarily serve to improve the learning process by generating distinguishable queries rather than being deployment candidates.
>
> Your intuition about exploration benefiting task performance is correct. In our experiments, particularly in the maze navigation task (Figure 3), the diverse policies' exploration does help discover better regions of the preference space. This knowledge is then transferred to the anchor policy through the reward model when diverse agents find trajectories that reveal hidden preferences or escape local optima, these discoveries are captured by the reward model through human feedback comparisons, which then guides the anchor policy toward better performance. In this way, all policies' exploration contributes to approximating the human-preferred task, and the exploration is indeed beneficial. Regarding combining policies: We would be interested to hear your thoughts on how you would envision combining them since it could be an interesting future direction ? We note that the current indirect transfer through reward model learning has proven effective in our experiments.
>
> **Q2 : Sensitivity to α**
>
> The parameter α controls when diverse agents receive diversity bonuses by setting a performance threshold relative to the anchor. Agents must achieve at least α times the anchor's return before diversity bonuses are applied, α defining a performance radius around the anchor policy.
>
> Higher α values (e.g., 0.9) require agents to stay close to the reward model's performance while exploring, keeping diverse behaviors within high-reward regions aligned with current preferences. Lower α values allow the diversity bonus even when an agent's performance is far from the anchor, enabling broader state space exploration but risking less informative queries that don't reflect learned preferences.
>
> We provide qualitative sensitivity analysis in Appendix Figure 8 for α ∈ {0.5, 0.7, 0.9}. Higher α values create stricter conditions, requiring stronger reward model alignment before diversity optimization kicks in. This results in trajectories targeting regions where both the reward model and diversity bonus are simultaneously high. We used α = 0.9 consistently across all environments without per-task tuning. Very low α values would let agents explore freely even with poor reward model performance, potentially generating uninformative queries.

---

> > ### Author Response · Authors · 2025-11-20
> >
> > **Q3: Discriminator training objective**
> >
> > The discriminator is trained to distinguish between all policies in the population (Equation 4)
> > This means the discriminator learns to identify which policy generated each trajectory, including identifying the anchor policy. Since the discriminator can recognize the anchor, it naturally provides diversity bonuses that push diverse policies away from anchor-like behaviors. However, crucially, only the diverse policies use these diversity bonuses in their optimization (Equation 3). The anchor policy optimizes purely for the reward model without any diversity term. This asymmetric design means: while the discriminator learns to distinguish all policies, only non-anchor policies are incentivized to behave distinctly from the anchor, which maintains its focus on exploitation.
> >
> > ---
> > Thank you for your feedback and efforts in reviewing our work. We look forward to hearing from you and providing any other clarifications during the discussion period.

---

### Author Response · Authors · 2025-12-01
**Discussion Summary**

We thank all reviewers for their constructive feedback and engagement during the rebuttal phase. The discussion period has been productive, with reviewers acknowledging that our responses successfully addressed their concerns.

**Recognition of Contributions**

All reviewers praised fundamental aspects of our work. Our population-based diversity mechanism was recognized as insightful and reasonably novel. Multiple reviewers highlighted excellent clarity and well-structured presentation. The similarity threshold mechanism was described as highly practical. Our experimental evaluation with clear performance gains under realistic feedback conditions was well received.

## Reviewer UaLC

**Initial Concerns** Clarifications needed on discriminator implementation, policy evaluation strategies, and α sensitivity.

**Resolution** We provided comprehensive explanations of our stability mechanisms (Appendix A.3.1). For α sensitivity, we explained how this parameter controls the performance threshold for diversity bonuses and provided qualitative analysis (Appendix Figure 8). We used α = 0.9 consistently across all environments without per-task tuning. We clarified how diverse agents benefit the shared reward model. These were primarily requests for clarification rather than fundamental methodological issues.

## Reviewer wJQR

**Initial Concerns** Questions about computational requirements, suggestions for additional related work discussion, minor notation clarifications, and baseline coverage.

**Resolution** We clarified that human feedback is the critical bottleneck, with our method reducing feedback requirements. We updated the manuscript with additional related work discussion including PPE and SENIOR (code unavailable for direct comparison, we cover similar uncertainty-based methods). We addressed minor typos and notation issues. The reviewer explicitly acknowledged our responses addressed their concerns and raised their score.

## Reviewer yKcr

**Initial Concerns** Questions about implementation details and computational requirements, overlapping confidence intervals in some DMControl results, and characterization of stability mechanisms.

**Resolution** We clarified the computational overhead is memory-based and manageable on consumer GPUs. We updated results with more seeds resulting in tighter confidence intervals. Our improvements are most pronounced in realistic scenarios with limited feedback and noisy evaluation. We demonstrated that stability mechanisms are principled design choices supported by ablation studies. We explained that 3 to 4 agents are enough to achieve optimal performance because the human feedback bottleneck limits how thinly feedback can be spread across the population. We clarified the distinction between state and preference space exploration. The reviewer noted most of their concerns have been addressed and raised their score by two points.

## Reviewer adkb

**Initial Concerns** Discussion of theoretical analysis, computational considerations, hyperparameter tuning across domains, and baseline comparison scope.

**Resolution** We clarified our contribution is empirical with comprehensive experimental validation. We explained computational overhead is manageable with parallelization, with human feedback being the primary bottleneck. We demonstrated hyperparameter robustness through our performance-constrained mechanism. We added RIME to both our discussion and experimental comparisons, and clarified differences with methods operating in fundamentally different settings.

## Updates Made During Rebuttal

We strengthened our empirical case with updated statistical analysis using more seeds and tighter confidence intervals. We expanded the related work discussion to include PPE and SENIOR. We added RIME as an additional baseline in experimental comparisons. We updated the manuscript to address minor notation issues and typos. We provided extensive clarifying discussions on implementation details, stability mechanisms, and design choices with supporting evidence from appendix materials. These updates directly addressed reviewer questions while strengthening manuscript quality and clarity.

**Rebuttal Outcomes**

Reviewer wJQR explicitly stated the rebuttal improved clarity and recognized the overall strength of our work, leading them to raise their score. Reviewer yKcr explicitly noted that most of their concerns have been addressed and raised their score by two points.

## Conclusion

All reviewer concerns were addressed. The issues raised were primarily requests for clarification and minor additional analysis rather than fundamental flaws. Our core contributions of population-based preference exploration, improved robustness to human inconsistency, and enhanced feedback efficiency remain intact and strengthened by the additional evidence. The reviewer acknowledgments and score improvements confirm the rebuttal adressed their concerns.

---

### Meta-Review · Area_Chair_EXk4 · 2026-01-05

**Summary:**

This paper introduces a population-based framework aimed at enhancing exploration in preference-based reinforcement learning. The paper addresses an important problem in preference-based RL, and the proposed idea is interesting. However, reviewers raised concerns regarding the adequacy of the baselines, the computational overhead, and the implementation complexity of the proposed approach. These concerns were not sufficiently addressed during the discussion period (see Reviewer Concerns). Therefore, I recommend rejection of this submission.

**Reviewer Concerns:**

### Reviewer UaLC

* [resolved] Implementation and experimental challenges with the discriminator

### Reviewer wJQR

* [remaining] Computational overhead

* [remaining] Comparison with PPE and SENIOR

* [resolved] Stronger Bayesian query selection/uncertainty-modeling baselines

### Reviewer yKcr

* [remaining] Implementation complexity and the number of hyperparameters are significantly increased.

* [resolved] Performance gain

###  Reviewer adkb

* [remaining] lack of theoretical analysis

* [remaining] computational overhead

**Reviewer Scores:**

* Reviewer UaLC: 6 $\rightarrow$ 6
* Reviewer wJQR: 2 $\rightarrow$ 4
* Reviewer yKcr: 4 $\rightarrow$ 6
*  Reviewer adkb: 4 $\rightarrow$ 4

---

### Decision · Program_Chairs · 2026-01-26

Reject